# A small molecule stabilizer rescues the surface expression of nearly all missense variants in a GPCR

**Taylor L. Mighell** [1] **& Ben Lehner** [1,2,3,4] ✉

Reduced protein abundance is the most frequent mechanism by which rare missense variants cause disease. A promising therapeutic avenue for treating reduced abundance variants is pharmacological chaperones (PCs, also known as correctors or stabilizers), small molecules that bind to and stabilize target proteins. PCs have been approved as clinical treatments for specific variants, but protein energetics suggest their effects might be much more general. To comprehensively assess PC efficacy for variation in a given protein, it is necessary to first assign the molecular mechanism explaining all pathogenic variants, then measure the response to the PC. Here we establish such a framework for the vasopressin 2 receptor (V2R), a G-protein-coupled receptor in which loss-of-function variants cause nephrogenic diabetes insipidus (NDI). Our data show that more than half of NDI variants are poorly expressed, highlighting loss of stability as the major pathogenic mechanism. Treatment with a PC rescues the expression of 87% of destabilized variants. The non-rescued variants identify the drug's predicted binding site. Our results provide proof-of-principle that small molecule binding can rescue destabilizing variants throughout a protein's structure. The application of this principle to other proteins should allow the development of effective therapies for many different rare diseases.

Rare genetic diseases pose a formidable challenge for global health. For any disease, the number of affected individuals is a small percentage of the population; however, in total, as many as 300 million people are affected by rare diseases[1]. Despite recent progress in computational methods[2], the identification of causal pathogenic variants and the determination of molecular mechanisms remains an arduous challenge[3,4]. Furthermore, developing effective therapies for genetic diseases for which only a small number of patients carry each causal variant is extremely challenging.

The most frequent mechanism by which missense variants cause rare diseases is reduced protein abundance. Large-scale experimental[5] and computational[6,7] surveys estimate that 40–60% of pathogenic variants are explained by loss of abundance. Compensating for this

reduced abundance therefore represents a potentially general strategy to treat rare diseases. PCs are typically small molecules whose binding increases the thermostability and subsequent steady-state expression level of a target protein. A striking example of PC success is in the treatment of cystic fibrosis, in which a combination treatment of two PCs and a channel potentiator offers effective treatment for people with the most common alteration, p.Phe508del, and some other variants[8]. PCs—also referred to as protein stabilizers or correctors—have also been developed for non-membrane proteins, including clinically approved PCs for lysosomal storage disease proteins[9] and the amyloidogenic transthyretin protein[10], as well as experimental PCs for the most frequently mutated tumor-suppressor protein, p53 (refs. 11,12). To maximize the effectiveness of PC therapy, however, the mechanism

[1]Centre for Genomic Regulation (CRG), The Barcelona Institute of Science and Technology, Barcelona, Spain. [2]Universitat Pompeu Fabra (UPF), Barcelona, Spain. [3]ICREA, Barcelona, Spain. [4]Wellcome Sanger Institute, Wellcome Genome Campus, Hinxton, United Kingdom. ✉e-mail: bl11@sanger.ac.uk

of all pathogenic variation for a given protein, as well as the response to PC, must be identified.

Depending on the mechanism of action, some PCs could have high specificity, rescuing only subsets of pathogenic variants localized in particular regions of the protein[13,14]. Alternatively, some PCs could behave simply, in accordance with the law of mass action, and have largely non-specific stabilizing effects that offset the destabilization by most variants in a protein, wherever they are located[12,15]. Most human proteins are marginally stable, meaning that only small changes in folding energy are required to produce large changes in folded protein abundance[16]. Indeed, in a collated set of 223 experimentally determined changes in Gibbs free energy of folding ($\Delta\Delta G$) values for membrane protein mutations, 91% were less than 3 kcal mol$^{-1}$ (ref. 17). Similarly, massive experimental mutagenesis of soluble proteins has confirmed that the vast majority of variants cause only small changes in folding energy[18], as do known pathogenic variants[5]. Such small changes in fold stability could potentially be easily compensated for by small-molecule binding, which can produce comparable changes in free energy to those induced by mutations[19]. Moreover, provided that free energies combine mostly additively[12,20,21], the stabilization conferred by small-molecule binding should be largely independent of the binding site and mutation location, as long as the compound specifically binds the native folded state.

The vasopressin 2 receptor (V2R) is a GPCR with an important role in water homeostasis in the kidneys[22]. The hormone arginine vasopressin (AVP) is the primary endogenous ligand of V2R, and AVP levels regulate the permeability of the kidney's collecting duct: elevated AVP levels lead to increased permeability, promoting water reabsorption. Upon AVP binding, V2R adopts an active conformation and couples primarily with $G_{\alpha s}$-containing heterotrimeric G proteins, leading to intracellular signaling that results in translocation of aquaporin-2 water channels to the plasma membrane[23]. When V2R function is lost, water reabsorption is compromised, leading to nephrogenic diabetes insipidus (NDI)[24]. Individuals with NDI experience chronic dehydration that can lead to severe clinical outcomes, and treatment options are available only to manage symptoms[25]. Highlighting the sensitivity and importance of this system, rare gain-of-function mutations that elevate the basal activity of V2R result in nephrogenic syndrome of inappropriate antidiuresis (NSIAD)[26]. The gene encoding V2R, *AVPR2*, is on the X chromosome. The majority of people with NDI are hemizygous males, although skewed X inactivation has been reported to cause NDI or subclinical phenotypes in some females[27]. Hundreds of *AVPR2* variants have been found in individuals with NDI, of which about half are missense variants; remainder are nonsense, small insertions or deletions or splice-site mutations[28]. Only a fraction of the missense variants has been experimentally characterized[23,28-35].

Here, we use V2R as a model system to directly test whether PCs can rescue all destabilizing mutants in a protein. First, we use a multiplexed assay to quantify the effects of all possible variants on the cell surface expression of V2R, revealing that more than half the known pathogenic variants strongly impair V2R expression, as do thousands of other missense variants throughout the protein. Strikingly, treatment with the V2R small-molecule binder tolvaptan rescues the expression of nearly all these variants. Only a small number are not compensated for, with these identifying functionally important sites and the drug binding site. The application of this approach and principle to other proteins should allow the development of general stabilizers for many genetic diseases.

## Results

### Massively parallel measurement of V2R surface expression

We used scalable and uniform nicking (SUNi) mutagenesis[36] to generate a saturation variant library containing all single amino acid changes to the V2R coding sequence. Mutagenesis primers were designed to

introduce a degenerate NNK or NNS codon at each position (depending on the wild-type codon, K = G or T, S = G or C), and random DNA barcodes were subsequently inserted into the plasmid backbone to enable identification of each variant with short-read sequencing. To link the full V2R variant sequence with the short DNA barcode, we used long read sequencing and successfully linked 66,031 barcode variants (Extended Data Fig. 1a). In total, 7,005 out 7,400 (94.7%) possible missense and nonsense variants were represented by at least 1 barcode, with a median of 5 barcodes per variant.

We implemented a fluorescence-activated cell sorting (FACS)-based approach to measure the surface expression of the library of variants in human cells[37,38]. First, the plasmid library was recombined into HEK293T landing-pad cells[39] (see Methods), ensuring that each cell carried exactly one variant. The V2R construct had a hemagglutinin (HA)-epitope tag at the amino terminus, which would be extracellular in a properly folded and trafficked receptor. Therefore, we performed immunostaining with a fluorescent antibody but without permeabilizing the cells, so that only receptors that had reached the membrane would contribute to signal. Then, we sorted cells into four bins, isolated DNA from the cells in each bin and used short-read sequencing to count the frequency of each variant across the bins (Fig. 1a).

Surface expression scores were calculated using the frequency of each variant in each bin multiplied by the geometric mean fluorescence value associated with each bin (see Methods). Across four replicates, we obtained high-confidence measurements for 6,844 (92.5% of possible; Methods) variants with high reproducibility (average pairwise replicate Pearson's $r = 0.90$, Extended Data Fig. 1b and data in Supplementary Table 1). Multiplexed scores show a strong correlation with FACS measurements from a series of isogenic cell lines containing single V2R variants ($r = 0.95$, Fig. 1i). Scores were normalized between a complete loss of function (designated as the median score of all nonsense variants in the first 300 positions of the receptor; assigned score 0), and the wild-type genotype (assigned a score of 1). Synonymous wild-type variant scores cluster distinctly from nonsense scores, and the distribution of all missense variants was bimodal: a large subset was expressed at near-wild-type levels, and a smaller subset near the level of nonsense variants (Fig. 1b). We used the top 95th percentile of truncation scores (0.35) and the bottom 95th percentile of synonymous wild-type scores (0.825) to categorize missense variants as well expressed (3,415 variants), moderately expressed (1,772 variants) or poorly expressed (1,025 variants, Fig. 1b).

A heatmap representation of the data reveals that nonsense mutations are uniformly damaging through all seven transmembrane (TM) helices, but mostly do not compromise surface expression in the unstructured, carboxy-terminal tail (Fig. 1c). TM helices are particularly intolerant to substitutions, especially to charged amino acids. In fact, substitutions in the TM helices as a population are significantly more damaging than are those in the extra- or intracellular loops (Mann–Whitney $U$ test, two-sided, $P = 3.16 \times 10^{-135}$, $P = 8.54 \times 10^{-153}$, respectively), whereas there is no difference between the extra- and intra-cellular loops ($P = 0.39$, Fig. 1d,g). Comparing the surface expression scores of all variants in each TM helix, variation in TM3 is the most damaging (median, 0.49; Fig. 1e). This is consistent with previous work suggesting that TM3 has a critical role in receptor stability[40] and acts as a 'structural hub'[41] for the TM bundle across class A GPCRs. TM2 is the second most mutation-sensitive (median, 0.56) TM helix. The predicted free-energy change ($\Delta G$) for membrane integration[42] of these helices indicates a relatively unfavorable process, with median predicted $\Delta G$ values of 3.1 and 3.7 for TM2 and TM3, respectively), suggesting that substitutions in these locations could further compromise an already inefficient process. By contrast, TM7 exhibits a median predicted $\Delta G$ of 4.2, indicating inefficient membrane incorporation; however, substitutions here are well tolerated (Fig. 1f).

TM regions are solvated in lipids and are therefore enriched in hydrophobic residues, compared with extra- or intracellular

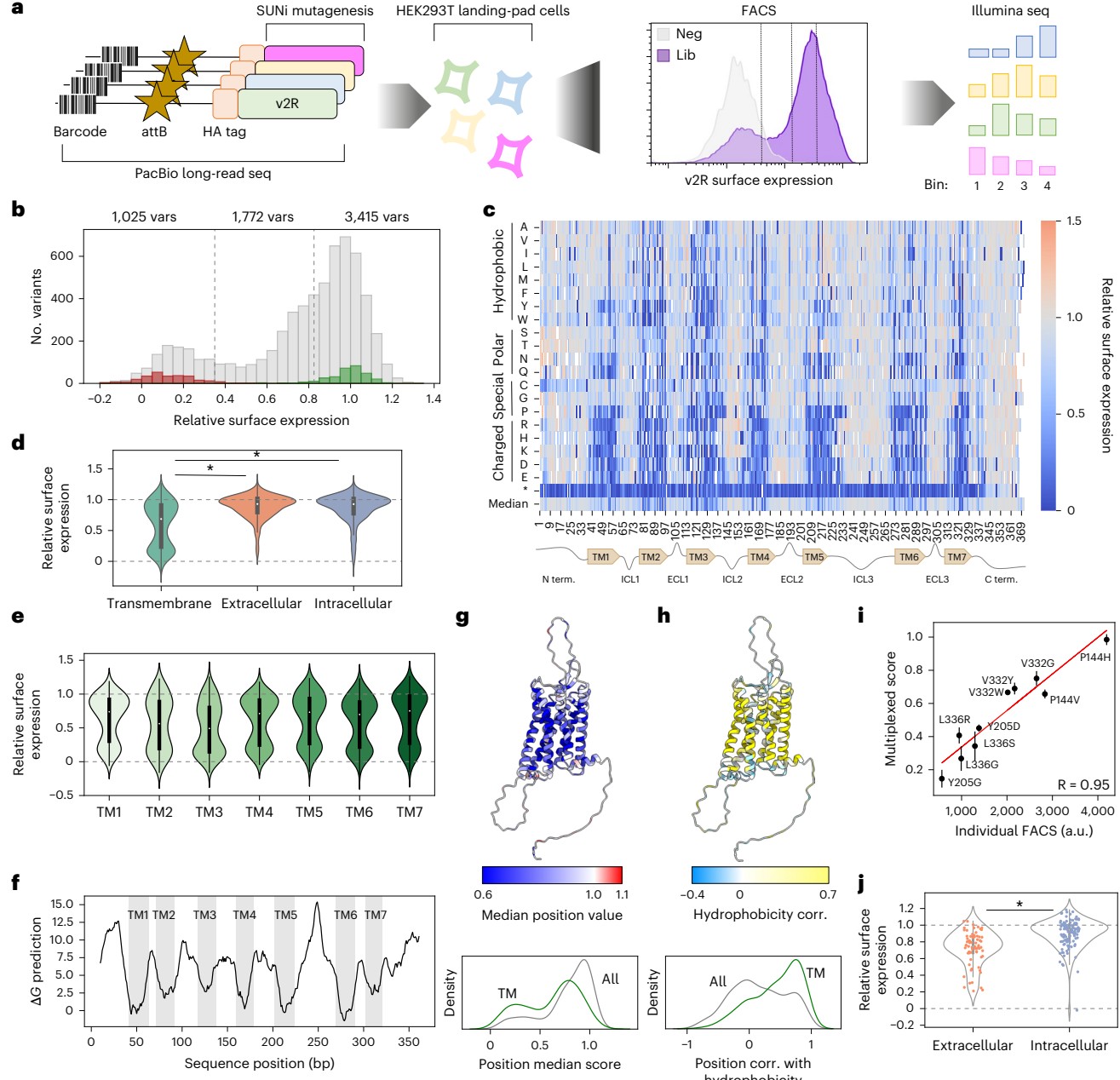

**Fig. 1 | Experimental approach and primary dataset overview. a**, A library of V2R variants was generated using SUNi mutagenesis and barcoded with random DNA barcodes. PacBio long-read sequencing was used to link variants with barcodes. Then, the library was integrated into HEK293T LLP-iCasp9-Blast landing-pad cells. Fluorescent antibody staining, followed by FACS, was used to fractionate cells by expression level, then DNA was sequenced with Illumina short reads to identify the variant abundances in each bin. Neg, negative control; Lib, library. **b**, Histogram of variant effect scores. Premature truncating variants are shown in red, synonymous wild-type variants are shown in green and missense variants are shown in gray. Dashed lines indicate the thresholds between well expressed, moderately expressed and poorly expressed variants. The number of missense variants in each category is reported above the graph. **c**, Heatmap representation of the data. V2R receptor positions are on the x axis, and mutant amino acid identities are on the y axis. Blue cells indicate surface expression below the wild-type level; gray indicates that surface expression is the same as the wild-type level; and red indicates that surface expression is above the wild-type level. **d**, Comparison of variant effects at positions in TM helices, or in extra- or intra-cellular loops. n = 2,620; 1,421; and 2,171 for TM, extracellular and intracellular, respectively. The center value (white circle) of each violin plot is the median; the box limits represent the first and third quartiles; and the whiskers extend to either the most extreme value, or interquartile range × 1.5, whichever is smaller. Mann–Whitney U test, two-sided, P = 3.16 × 10⁻¹³⁵, P = 8.54 × 10⁻¹⁵³. **e**, Comparison of variant effects at positions in TM helices 1–7. Violin plot features are the same as in **d**. **f**, Predicted ΔG of membrane insertion (ΔG prediction server v1.0; full protein scan; helix length, 21). **g**, Top, median position surface expression score displayed in color on the AlphaFold2-predicted V2R structure. Bottom, density plot of median position scores for the whole receptor (gray) or in TM helices (green). **h**, Top, per-position correlation of surface expression scores with Kyte–Doolittle hydrophobicity shown in color on the AlphaFold2-predicted V2R structure. Bottom, density plot of per-position correlations for the whole receptor (gray) or within TM helices (green). **i**, Comparison of ten variant scores measured individually or en masse (DMS measurements). The s.e.m. is shown for DMS measurements. The linear fit is shown in red. Measurement of isogenic cell lines was done once but encompasses measurements from >10,000 cells for each genotype. a.u., arbitrary units. **j**, Effects of substitutions with cysteine in the extra- or intra-cellular portions of the receptor. n = 69, 116 for extracellular and intracellular, respectively. The violin features are the same as in **d**. Mann–Whitney U test, two-sided, P = 7.3 × 10⁻¹⁰.

residues. Substitutions in the TM regions that decrease hydrophobicity have negative effects on surface expression (Spearman's $\rho = 0.28$, $P = 1.8 \times 10^{-48}$), whereas the relationship is much weaker and in the opposite direction for the extra- or intracellular regions ($\rho = -0.05$, $P = 0.002$, Supplementary Fig. 1c). Visualization of each residue's preference for hydrophobicity (calculated as rank correlation of surface expression scores with Kyte–Doolittle hydrophobicity) agrees with this notion (Fig. 1h), but highlights several residues in the core of the receptor that prefer hydrophilic residues (Extended Data Fig. 1f).

V2R undergoes post-translational modification in the form of O- and N-glycosylation on the N-terminal tail[43] as well as palmitoylation on the C-terminal tail.[44] Positions 22–24 represent the N-glycosylation motif (N-X-S/T), and substitutions at positions 22 and 24 are much more deleterious than are their neighbors in the unstructured N terminus (respective median position score, 0.77 and 0.68; Extended Data Fig. 1d). Whereas O-glycosylation was reported at several serines and threonines in the N terminus[43], we see strong mutational effects only at S5 and T6 (respective median position scores, 0.74 and 0.73), suggesting these are the only critical glycosylation sites, at least in HEK cells with this overexpression system. Substitutions to cysteine are significantly more deleterious in the extracellular loops than in the intracellular loops (Fig. 1j, Mann–Whitney $U$ test, two-sided, $P = 7.3 \times 10^{-10}$). The extracellular cysteine substitutions likely disrupt a conserved disulfide bond[41]. Finally, palmitoylation at cysteines 341 and 342 have been reported to be important for surface expression[44]; our data show that in HEK cells, substitutions at these sites are well tolerated (Extended Data Fig. 1e).

### The contribution of V2R surface expression to pathogenicity

We next sought to understand the contribution of surface expression defects to V2R-related disease. To do this, we collated clinical variants from ClinVar[45], population variants from gnomAD[29] and variants reported in individuals with NDI or NSIAD in the Human Gene Mutation Database[46]. There are striking differences between the surface expression scores for putatively benign variants (in gnomAD, or benign or likely benign in ClinVar) and the putatively loss-of-function ones (pathogenic or likely pathogenic in Clinvar, or NDI, Fig. 2a). There are no poorly expressed variants in the gnomAD, benign or likely benign sets, and only 30.7% and 26.6% are moderately expressed in likely benign and gnomAD, respectively (Fig. 2b). By contrast, for pathogenic, likely pathogenic and NDI variants, 40%, 53.3% and 55.6%, respectively, are poorly expressed, whereas 26.6%, 26.6% and 30%, respectively, are moderately expressed. There are only five known NSIAD variants, and they are constitutively active[47]. Four are moderately expressed and one is well expressed, in accordance with the gain-of-function mechanism. Among variants of uncertain significance, 15.6% and 25% are poorly and moderately expressed, respectively (Fig. 2b). Stratifying gnomAD variants by allele frequency shows that common variants (allele frequency > 0.001) are all well expressed (Fig. 2c).

### Classifying pathogenic variant mechanisms

Although computational variant effect predictors (VEPs) can distinguish pathogenic from benign variation with increasing accuracy, they cannot perform the crucial task of determining molecular mechanisms for pathogenicity (Fig. 2d). First, we evaluated how well computational VEPs can discriminate pathogenic (NDI alleles) from putatively benign variation (gnomAD alleles). ESM1b[48], a protein language model, achieves an area under the receiver operator characteristic curve (AUROC) of 0.91. EVE[49], which uses multiple sequence alignments to model evolution, achieves an AUROC of 0.92. Finally, AlphaMissense[50], which uses structural predictions, population variant frequencies and evolutionary data, achieves an AUROC of 0.94. In comparison, surface expression scores achieve an AUROC of 0.84, again suggesting that surface expression is a major determinant of pathogenicity.

However, models designed to predict effects of variation on protein stability performed less well than did the empirical surface expression scores. RaSP[51] achieved an AUROC of 0.76, and ThermoMPNN[52] achieved one of 0.80 (Fig. 2e). We also compared the predictors with the surface expression scores, and with each other. The VEPs correlated better with the surface expression data, with $\rho = 0.6$, 0.57 and 0.56 for AlphaMissense, EVE and ESM1b, respectively (Fig. 2f and Extended Data Fig. 2a–c). ThermoMPNN and RaSP correlated less well, at 0.46 and 0.33, respectively (Fig. 2f and Extended Data Fig. 2d,e). The poorer correlation of the stability predictors likely reflects the difficulty of predicting stability changes for membrane proteins, especially given that RaSP and ThermoMPNN were trained on soluble proteins. Finally, we explored whether mutation effect predictors could capture the pathogenicity of the NSIAD gain-of-function mutations. The number of variants is too small to make strong conclusions, but there seems to be much variability for the different models' predictions, suggesting that gain-of-function variants could be more difficult to predict than loss-of-function ones (Extended Data Fig. 2a–e).

We hypothesized that our surface expression scores could clarify the mechanisms behind pathogenic variants, specifically whether a V2R variant is pathogenic owing to decreased surface expression or an impaired signaling capability. We posited that variation at positions close to the natural ligand (AVP) binding site in 3D space are more likely to disrupt signaling than is variation located further away. Indeed, we found that well and moderately expressed NDI variants are closer to AVP in a solved structure (Mann–Whitney $U$ test, two-sided, $P = 1.5 \times 10^{-3}$ and $7.7 \times 10^{-4}$ for well-expressed and moderately expressed variants compared with poorly expressed variants; PDB structure 7KH0; Fig. 2g,h).

Although there are more than 100 known NDI variants, this is likely only a small fraction of pathogenic variation in V2R. Therefore, we used AlphaMissense to predict all pathogenic variants in V2R (AlphaMissense threshold, 0.564 (ref. 50)). There are 2,911 variants predicted to be pathogenic, of which there are high-confidence surface expression scores for 2,595. Of these, 979 (37.7%) are poorly expressed, 953 (36.7%) are moderately expressed and 663 (25.5%) are well expressed (Fig. 2i).

---

**Fig. 2 | V2R clinical genetics and variant effect predictors. a**, Surface expression scores of V2R variants in gnomAD, ClinVar categories or Human Gene Mutation Database (HGMD) categories. Patho., pathogenic. **b**, Fraction of each variant category that is poorly, moderately or well expressed. **c**, Relationship between allele frequency in gnomAD and surface expression score. The dashed line indicates an allele frequency of 0.001, a commonly used threshold for common versus rare variation. **d**, Although VEPs are quite accurate for predicting pathogenicity, they cannot inform on the molecular mechanism of pathogenicity, highlighting the importance of mechanistic screens. **e**, Receiver operating characteristic curve for different VEPS, or the empirical surface expression scores, for distinguishing NDI alleles from gnomAD alleles. AUC, area under the curve. **f**, Correlations between the predictors and the empirical surface expression scores. $P < 1 \times 10^{100}$ in all cases. **g**, Distance between the endogenous ligand (AVP) and NDI variants with different expression levels. $n = 26$, 26 and 69 for well expressed, moderately expressed and poorly expressed, respectively. The center value (white circle) of each violin plot is the median; the box limits represent the first and third quartiles; and the whiskers extend to either the most extreme value, or interquartile range ×1.5, whichever is smaller. Mann–Whitney $U$ test, two-sided, $P = 1.5 \times 10^{-3}$ and $7.7 \times 10^{-4}$ for well and moderately expressed versus poorly expressed. **h**, NDI variants of different expression levels illustrated on the V2R structure (PDB ID: 7KH0). **i**, Surface expression of all variants predicted by AlphaMissense to be pathogenic (gray histogram), along with the distribution of all missense variants (black line). $P = 2.6 \times 10^{-5}$. **j**, Distance between variants with different expression levels to AVP in the solved V2R structure. $n = 654$, 946 and 977 for well expressed, moderately expressed and poorly expressed, respectively. $P = 2.2 \times 10^{-21}$ for well versus poorly expressed; 0.03 for moderately versus poorly expressed; $1.9 \times 10^{-9}$ for well versus moderately expressed; Mann–Whitney $U$ test, two-sided. Violin plot features are the same as in **g**.

AlphaMissense predicted pathogenic variants have bimodal surface expression scores, just like the distribution of all missense variants. However, although the mode of the higher expression peak for all missense variants is near 1.0, the mode of the higher expression peak for

AlphaMissense pathogenic variants is around 0.8; this is likely because the majority of them are in TM helices (1,789 out of 2,595, 68.9%). The mode of the higher expression peak for TM variants is also near 0.8 (see Fig. 1d). Finally, we compared the proximity of AlphaMissense

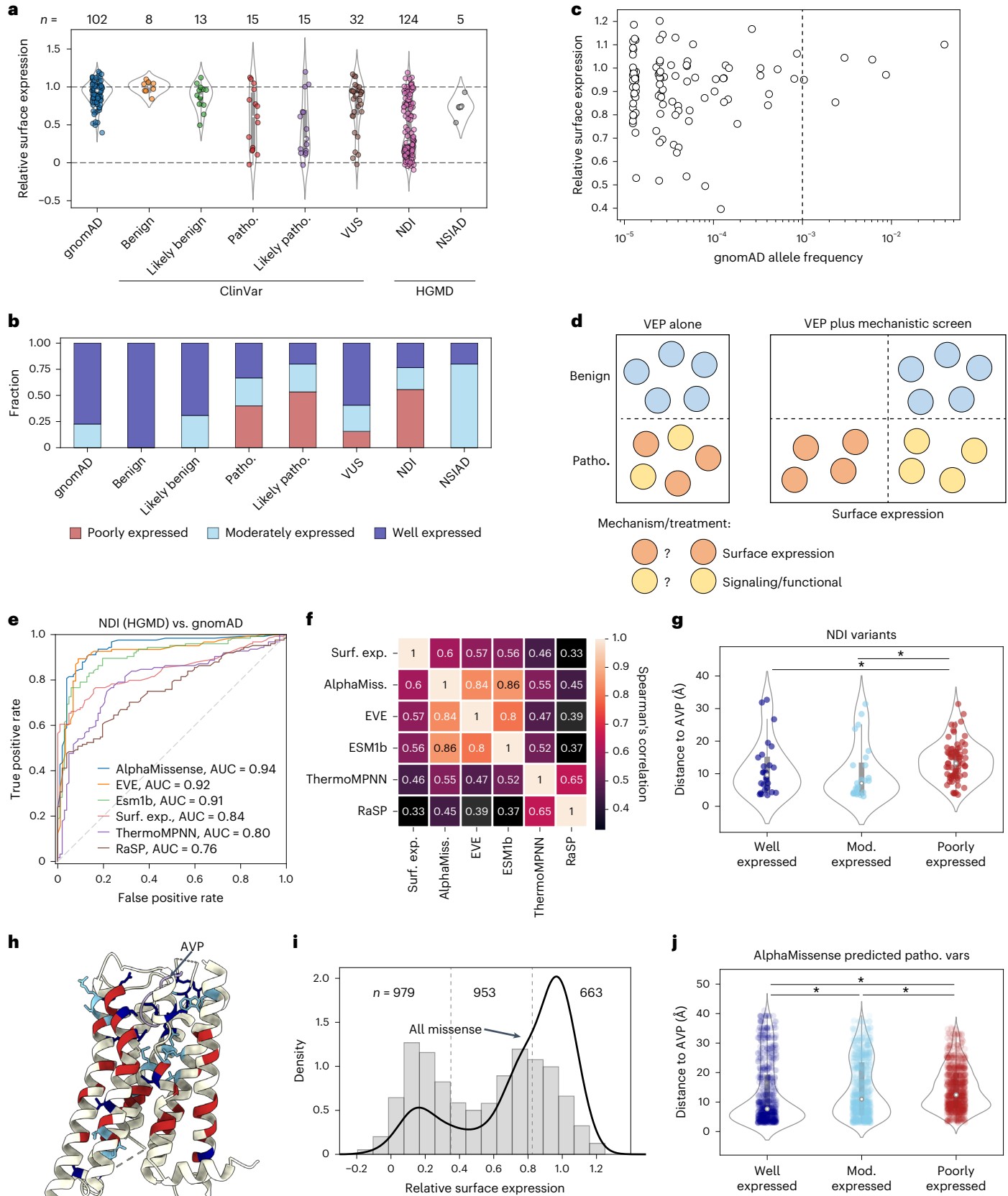

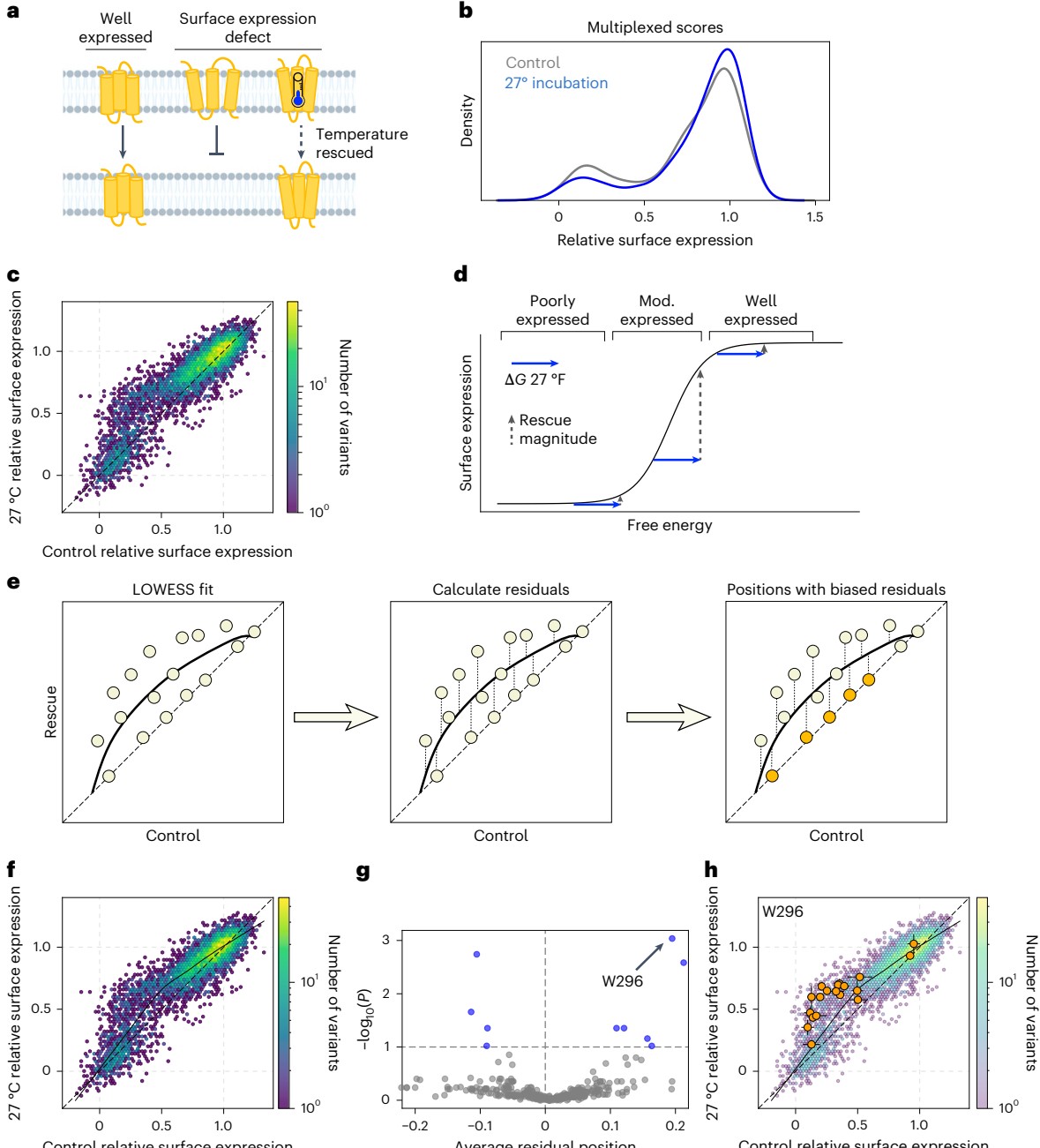

**Fig. 3 | Thermodynamic rescue of V2R variants. a**, Well-folded receptors are properly trafficked to the membrane, but folding-defective variants are not properly trafficked. These experiments seek to test which variants can be rescued by thermodynamic rescue. **b**, Multiplexed score distribution for surface expression of all missense variants in control (37 °C) and rescue (27 °C) conditions. **c**, Comparison of all missense variants in control and rescue conditions. **d**, The relationship between free energy change and surface expression. The relationship between these two values is sigmoidal, meaning that the same free energy change (for example, that conferred by reducing culture temperature from 37 °C to 27 °C) can have effects of different magnitude on surface expression. **e**, The strategy used to identify sites with higher or lower

levels of rescue than expected. A LOWESS curve is fit to the control versus rescue data; residuals to this line are calculated for all variants; positions with biased residuals are identified. **f**, Control versus rescue surface expression; LOWESS fit in solid black; the dashed $x = y$ line is shown. **g**, Volcano plot comparing the average residual of all positions with the FDR adjusted (FDR = 0.1); $-\log_{10}(P)$ was derived from Mann–Whitney $U$ tests comparing the variant residuals at each position with all variant residuals at all other positions. The position with the most significantly different set of residuals is highlighted (W296). **h**, All missense variant effects at W296 are plotted with the whole population of missense variants in the background. Error bars represent the s.e.m.

pathogenic variants with AVP in the solved V2R structure. As with the NDI variants, the well-expressed and moderately expressed pathogenic variants are closer to AVP in 3D space than are the poorly expressed variants ($P = 2.2 \times 10^{-21}$ and 0.03, respectively, Mann–Whitney $U$ test, two-sided, Fig. 2j), consistent with a specific signaling defect for variants that have at least moderate expression level. For this much larger

set, the well-expressed variants are also closer to AVP than are the moderately expressed ones ($P = 1.9 \times 10^{-9}$, Fig. 2j).

## Temperature rescue of V2R variants
In principle, variants that are poorly expressed owing to decreased thermodynamic stability should be rescued by incubating the cells

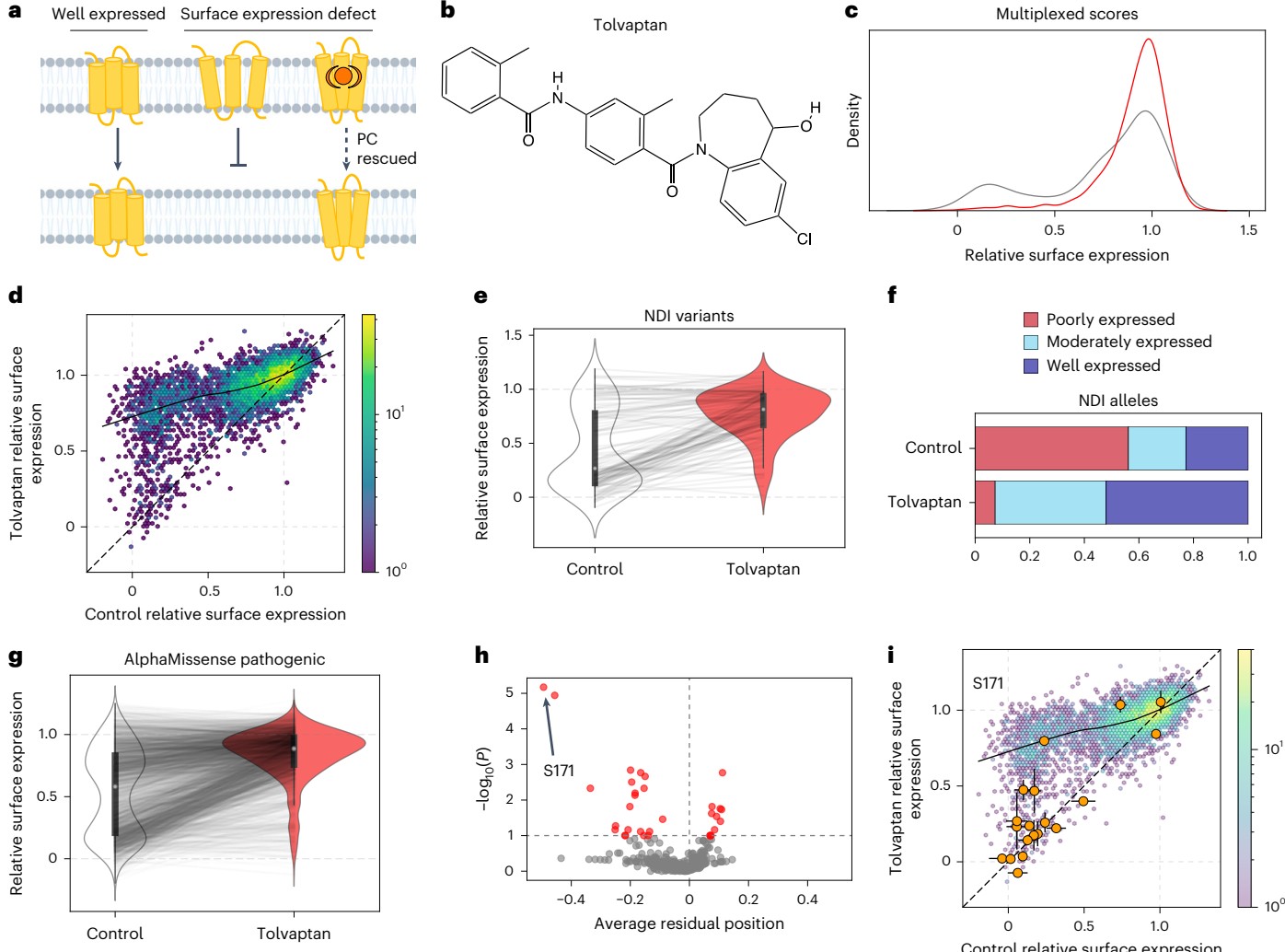

**Fig. 4 | Pharmacological rescue of V2R variants. a**, Well-folded receptors are properly trafficked to the membrane, but folding-defective variants are not properly trafficked. These experiments seek to test which variants can be rescued by pharmacological rescue. **b**, The chemical structure of tolvaptan. **c**, Surface expression multiplexed score distribution of all missense variants in control (DMSO) and rescue (tolvaptan) conditions. **d**, Comparison of all missense variants in control and tolvaptan conditions; the solid LOWESS fit line and dashed x = y line are shown. **e**, NDI variants surface expression in control compared with the tolvaptan condition. n = 123. The center value (white circle) of each violin plot is the median; the box represents the first and third quartile; and the whiskers extend to either the most extreme value, or interquartile range × 1.5, whichever is smaller. **f**, The fraction of NDI variants that are poorly, moderately or well expressed in the control and tolvaptan conditions. **g**, Surface expression of AlphaMissense-predicted pathogenic variants in the control versus tolvaptan condition. n = 2,555. Violin plot features same as in **e**. **h**, Comparison of the average residual of all positions with the FDR-adjusted (FDR = 0.1) −$\log_{10}(P)$ derived from Mann–Whitney *U* tests comparing the variant residuals at each position with all variant residuals at all other positions. The position with the most significantly different set of residuals is highlighted (S171). **i**, All missense variant effects at S171 are plotted with the whole population of missense variants in the background. Error bars represent the s.e.m.

expressing the variant library at reduced temperature (Fig. 3a). Although reduced temperature could also affect biosynthetic processes that result in increased expression[53], we assume that, in the majority of the cases, rescue is through thermodynamic stabilization[14]. We sought to understand what fraction of V2R variants could be rescued by temperature reduction by culturing the cells expressing the V2R variant library at 27 °C. After 24 h at 27 °C, FACS analysis revealed a shift in the distribution of surface expression, with more cells with high V2R expression levels (Extended Data Fig. 3a). To gather quantitative data for all variants, we then sorted and sequenced the libraries rescued at 27 °C. High-confidence measurements were collected for 6,787 missense and nonsense variants (91.7% of possible), and replicates were highly correlated (r = 0.91, Extended Data Fig. 3c and data in Supplementary Table 1). The distribution of multiplexed missense variant scores mirrors the FACS data, with a shift of some variants from the low-expressing to the high-expressing peak (Fig. 3b).

Then, we compared the surface expression of all variants at 27 °C compared with their expression under the control condition, 37 °C. The magnitude of rescue is greatest for variants with a moderate expression level (Fig. 3c). However, variants that are well or poorly expressed in the control condition do not show large changes. This result is consistent with temperature reduction having a constant, additive change in free energy, which, in turn, has a non-linear effect on the phenotype of interest[16,54], in this case steady-state expression level. Namely, as has been observed before in similar experimental paradigms[55,56], the magnitude of rescue is greatest for variants whose free energy values begin in the steepest part of the curve of the free-energy expression level (Fig. 3d).

To explore the generality of rescue achieved through temperature reduction, we designed an approach to identify positions with varying levels of rescue compared with all other positions. First, we fit a LOWESS curve to the control versus rescue data, establishing a null expectation

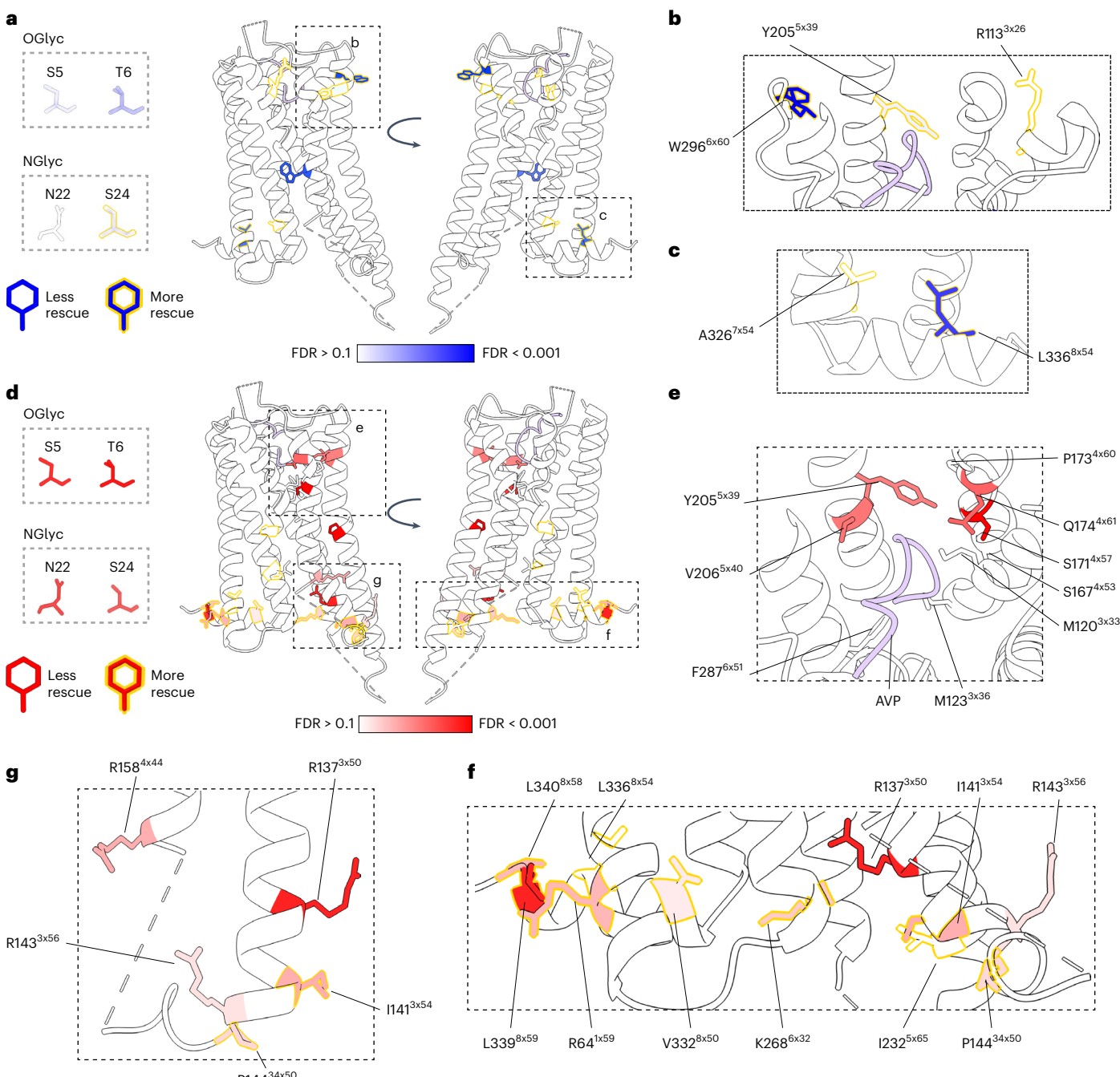

**Fig. 5 | Identifying sites with more or less rescue than expected. a–c**, Sites that are rescued more (blue with yellow outline) or less (blue with no outline) than expected by 27 °C. Glycosylation sites are not resolved in the crystal structure, so these are shown individually on the left. Sites of interest outlined with dashed boxes on the structure are shown to the right. **d–g**, Sites that are rescued more (red with yellow outline) or less (red with no outline) than expected by tolvaptan. Sites of interest outlined with dashed boxes on the structure are shown to the right and below. OGlc, O-glycosylation; NGlc, N-glycosylation.

for the magnitude of rescue at each control surface expression value. Following this, we calculate residuals for each variant relative to the fit line. Then, the residuals for each position are compared with those at all other positions to find outliers (Fig. 3e). After fitting the LOWESS curve, we used a two-sided Mann–Whitney $U$ test to identify positions with significantly biased residuals (Methods and Supplementary Table 2). For the 27 °C condition, out of 371 positions, only 4 positions are rescued less than expected; 6 are rescued more than expected on the basis of the model, (false discovery rate (FDR) = 0.1, Fig. 3g,h). Among these ten positions are three glycosylation sites (positions 5, 6 and 24), which would be expected to have effects beyond simple

thermodynamic destabilization. This suggests that the majority of variants can be rescued by temperature reduction.

## Pharmacological chaperone rescue of V2R variants

On the basis of the broad effectiveness of temperature rescue, we predicted that PC binding could have a general rescue effect. PCs are typically hydrophobic small molecules that cross the plasma membrane and stabilize a target protein by binding to the folded state to promote trafficking (Fig. 4a). Although PCs have effectively corrected NDI-associated V2R variants both in vitro and in the clinic[57], uncertainty around how general the effect of any PC would be has

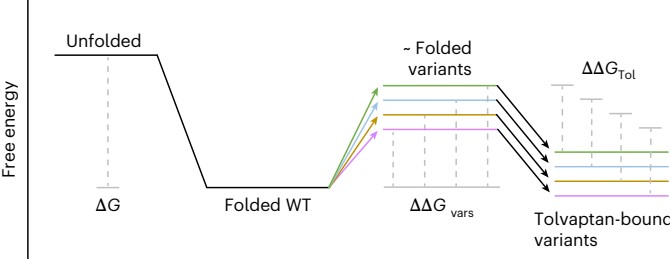

**Fig. 6 | Additive model of tolvaptan rescue.** Free energy model of V2R folding, genetic variation and rescue. We propose a model in which variants contribute different $\Delta\Delta G$ effects to the system, and tolvaptan contributes a constant $\Delta\Delta G$. In most cases, the $\Delta\Delta G$ of variants ($\Delta\Delta G_{var}$) and $\Delta\Delta G$ ($\Delta\Delta G_{Tol}$) of tolvaptan combine additively.

been a major obstacle[23]. PCs might have quite specific effects, such that each PC rescues only a subset of structurally related variants[13,14]. Alternatively, PCs might generally rescue mutants with reduced thermodynamic stability[12]. Tolvaptan (also known as OPC-41061) is a V2R-specific, competitive, small-molecule antagonist[58] (Fig. 4b), approved for treating autosomal dominant polycystic kidney disease[59], in which AVP-V2R signaling is misregulated. However, tolvaptan has been explored in vitro for its activity as a PC. In most cases, tolvaptan not only rescues surface expression, but also enables some level of AVP-mediated signaling[33,35,60,61]. However, as for PCs in general, tolvaptan has been tested for effectiveness on only a very limited set of variants (<15)[33,35,60,61].

Therefore, we treated cells bearing the V2R variant library with tolvaptan for 24 h and then profiled the population surface expression with FACS. Compared with the 27 °C condition, the tolvaptan condition has a stronger rescue effect, with more cells shifting from low to high expression (11% of cells shifting in tolvaptan compared with 5.3% at 27 °C, Extended Data Fig. 3b).

We sorted and sequenced the tolvaptan-rescued libraries and collected high-confidence measurements for 6,759 missense and nonsense variants (91.3% of possible, data in Supplementary Table 1). Replicates were well correlated ($r = 0.74$, Extended Data Fig. 3d), and multiplexed measurements were consistent with individual variant measurements in isogenic cell lines ($r = 0.85$, Extended Data Fig. 3e). Multiplexed missense score distributions match the FACS distributions and emphasize a near-complete shift of variants from low to high expression (Fig. 4c). Likewise, directly comparing surface expression levels in the presence and absence of tolvaptan indicates nearly universal rescue (Fig. 4d).

Next, we assessed the efficacy of tolvaptan rescue for known and predicted pathogenic variants. Tolvaptan is remarkably effective at rescuing NDI alleles: of 69 poorly expressed variants in the control condition, 60 (87.0%) are rescued to at least moderately expressed levels (Fig. 4e,f). We next evaluated effectiveness on variants predicted to be pathogenic by AlphaMissense. Strikingly, 835 of 965 poorly expressed variants (86.5%) are rescued to at least moderately expressed levels (Fig. 4g and Extended Data Fig. 3f). Tolvaptan has minor effects on NSIAD variant expression, likely because these are moderately or well expressed in the control condition (Extended Data Fig. 3g). As an alternative estimation of the extent of rescue, we identified those variants with tolvaptan rescue magnitude (that is, tolvaptan expression – control expression) greater than the 95% confidence interval of tolvaptan expression. Although the expected rescue magnitude for well-expressed variants is small, when considering variants with low control expression (<0.6), 88.4% are significantly rescued (Extended Data Fig. 3h).

To further investigate tolvaptan rescue, we fitted a LOWESS curve (Fig. 4d) and analyzed the residuals to this line to identify positions

exhibiting greater or lesser rescue levels than expected (Supplementary Table 2). More positions have specific interactions with tolvaptan than with 27 °C (32 compared with 10); 21 positions are rescued less than expected, and 11 are rescued more than expected (Fig. 4h,i, FDR = 0.1). Of the ten outlier residues for 27 °C rescue, six are also outliers for tolvaptan: S5, T6 and S167[4x53] are below expected levels in both cases, and L336[8x54] (GPCRdb[62] numbering in superscript) is rescued more than expected levels in both cases. Notably, although S24 and and Y205[5x39] are rescued more than expected at 27 °C, they are actually less rescued than expected in the tolvaptan condition. Overall, however, rescue in the two conditions is correlated ($\rho = 0.49$, Supplementary Fig. 3i).

### Identifying drug binding and functional sites
Next, we sought to understand the structural features associated with rescue outlier positions. For 27 °C rescue, three out of ten outlier positions are glycosylation sites. Both O-glycosylation sites (S5 and T6) are rescued less than expected, but S24 is actually rescued more than expected, suggesting that temperature reduction can compensate the N-glycosylation defect (Fig. 5a, residues colored according to FDR-adjusted P value, residues outlined in yellow are rescued more than expected, those not outlined are rescued less than expected). Of the others, three are near the orthosteric site (Fig. 5b) and two are in or near helix 8 (Fig. 5c).

For tolvaptan, all four glycosylation sites (S5, T6, N22 and S24) are rescued less than expected (Fig. 5d), implying that tolvaptan cannot compensate for the lack of proper glycosylation. In addition, rescue for a cluster of residues on one side of the orthosteric binding site is worse than expected (Fig. 5e). We suspected that this could be the tolvaptan-binding site. Indeed, a study employing molecular dynamics and site-directed mutagenesis[63] determined that the most important residues for tolvaptan antagonism are M123[3x36], F178[3x36], Y205[5x39], V206[5x40], and F287[6x51], of which all but F178[3x36] are rescued less than expected by tolvaptan. Finally, R137[3x50], in the highly conserved E/DRY motif, and two arginines in close proximity (R158[4x44] and R143[3x56]) are rescued less well than expected (Fig. 5f). The E/DRY motif is well conserved among class A GPCRs and has an important role in stabilizing the inactive state; mutations in this motif cause constitutive activity in various GPCRs[64]; similarly mutations at R137[3x50] in V2R are known to cause constitutive signaling activity[47], so this cluster of substitutions might render tolvaptan less effective by biasing the receptor to the active conformation. An examination of the sites at which variants are rescued more than expected highlights a cluster of residues at the intracellular interface of the receptor (Fig. 5g). Substitutions at these sites could potentially affect signaling and/or internalization; PC stabilization of the inactive state might therefore have an exaggerated effect here. Further mechanistic studies would be required to understand the behavior of substitutions at these sites.

Taken together, these results demonstrate nearly universal surface expression rescue of destabilized V2R variants with the PC tolvaptan. The few sites at which variants are consistently not rescued shed light on the likely tolvaptan-binding site, as well as other functional sites of the receptor.

## Discussion
Reduced expression due to impaired fold stability is the predominant mechanism by which missense variants cause disease, including in both soluble[5,65–68] and membrane proteins[69–72]. The destabilization conferred by most missense variants in most proteins is, however, small, suggesting that re-stabilization of a protein by a small-molecule PC that binds the native state might confer sufficient free energy to rescue the effects of very diverse substitutions throughout a protein's structure. For this approach to be effective, however, the mechanism for all pathogenic variants in a protein, as well as their response to PC, must be prospectively classified. Here, we have implemented this framework and tested the 'universal PC' hypothesis for V2R. First, we show that more than

half of known loss-of-function variants are poorly expressed, and use temperature reduction to show that the vast majority of variants lose expression as a result of thermodynamic instability. On this basis, we test the efficacy of tolvaptan, a well-characterized antagonist and PC of V2R. Our results show that binding of a single small-molecule PC can rescue the cell surface expression of nearly all variants throughout the receptor's structure.

Previous low-throughput studies mostly tested the efficacy of PCs on a limited number of variants, and the failed rescue of individual variants lead to suggestions of widespread idiosyncratic effects between PCs and variants[32–35]. More systematic efforts testing hundreds of variants in rhodopsin[14,55] and CFTR[56] demonstrated rescue of many variants, but the authors also suggested substantial region-specific differences in rescue. The rhodopsin investigations emphasized differences in rescue between mutations in transmembrane helices 2 and 7 (ref. [14]), but subsequently found that 67 out of 69 reduced expression retinopathy variants had measurably increased expression in the presence of a PC[55], consistent with our findings. CFTR has multiple folding domains[73], and PCs were found to be most effective for variants in close proximity to the PC-binding site[56]. Class A GPCRs then, as single-domain proteins, could be amenable targets for PCs. Here, we have profiled an order of magnitude more variants than previous studies, demonstrating protein-wide, nearly universal PC rescue of variants.

We expect that our results will generalize to other proteins and PCs, on the basis of empirical and theoretical evidence. Most proteins are marginally stable, meaning that large changes in expression result from substitutions with $\Delta\Delta G$ of only a few kcal mol$^{-1}$ (ref. [16]), and indeed large-scale surveys of mutational effects show most pathogenic substitutions cause only small changes in fold stability[5,17,18]. It is reasonable to think that such small changes in fold stability can be easily compensated for by small-molecule binding in many proteins. Tolvaptan binding to V2R, for example, confers a free energy change of approximately −12 kcal mol$^{-1}$ (ref. [63]), potentially completely offsetting the destabilizing effects of nearly all variants, provided that drug concentrations are high enough.

Accordingly, small molecules could offer a much more general strategy to rescue poorly expressed variants than is commonly perceived. The law of mass action means that, to rescue expression, a small molecule must specifically bind to the folded conformation of a protein, with sufficient energy to shift the folding equilibrium toward the folded state (Fig. 6). There is no requirement to specifically recognize the mutated amino acid or to bind any particular region of the protein. Similar principles apply when multiple mutations are combined in a protein, where large-scale experimental analyses have revealed that changes in fold stability are nearly always additive when diverse mutations are combined in a single protein[21,74,75]. Mutations, like small-molecule binding, can be viewed as perturbations to a protein that lead to changes in free energy. We believe, therefore, that many small molecules binding to proteins with sufficient free energy will behave as general or universal PCs.

In further support of the general, energetically additive view of PC efficacy, the small number of variants that are not rescued by tolvaptan mostly have clear mechanistic explanations: substitutions in the binding site of tolvaptan interfere with binding of the drug, and variants that affect expression by mechanisms other than reduced fold stability are not rescued, here exemplified by those at post-translational modification sites. The variants that are not rescued by a PC can be rapidly identified by selection and sequencing experiments and excluded from clinical trials. This approach to identifying changes in abundance that are not rescued by small-molecule binding is a potentially very general strategy to rapidly identify drug-binding sites in proteins.

High-throughput protein abundance selection assays have now been developed for many different protein classes[18,76,77]. Similar to the present study, these assays could quickly assess the efficacy of PCs across all variants in a protein, to prioritize broadly effective PCs.

One limitation of our tolvaptan-rescue finding is that we have not established the fraction of rescued variants that can signal at the membrane. However, there is reason to expect that a significant portion will retain signaling activity. Several studies have demonstrated detectable signaling for PC-rescued V2R[33,35,60,61] as well as for other GPCR variants in vitro and in vivo[78,79]. Indeed, a small clinical trial found that another V2R antagonist PC, SR49059, exhibited clinical improvement in people, although the trial was discontinued owing to off-target effects.

The demonstration that a PC rescues the expression of most missense variants throughout the structure of a protein has wide-ranging implications for rare disease research. Previous experimental[5] and computational[6,7] approaches estimate that 40–60% of pathogenic variants are explained by loss of stability or abundance (which is in line with our findings here), suggesting a broad scope for PC therapy. Such general PCs will not have to bind to specific sites in a protein, and they will not need to be tailored to each pathogenic variant. Rather, in accordance with the simple principle of additive free energies, any molecule that binds specifically to the folded state of a protein with sufficient free energy is a potential universal PC.

## Online content

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

## Methods

### V2R saturation mutagenesis and barcoding

The sequence for the barcoded attB-HA-V2R construct is available as a genbank file (https://zenodo.org/records/14216036). To introduce all single amino acid substitutions, we used SUNi mutagenesis[36]. In short, oligonucleotides were designed to encode all amino acid substitutions (using NNK or NNS degenerate codons) with flanking sequence of 18–40 bases to optimize melting temperature and ensure the presence of a 5′ GC clamp[36]. Oligonucleotides were ordered as an oPool from Integrated DNA Technologies (IDT, Supplementary Table 2). Four microliters of final product from the SUNi protocol was electroporated into 50 µl of 10β high-efficiency electrocompetent *Escherichia coli* cells (New England Biolabs (NEB)), followed by 1 h recovery with 2 ml of super optimal broth with catabolite repression (SOC) medium. At this point, 0.5% of the recovery was plated onto an LB agar plate with ampicillin, and 99.5% was inoculated into 100 ml of LB liquid with ampicillin. The total transformant number, estimated by the number of colonies on the plate, was estimated to be >200,000. The plasmid was isolated the following morning using the Qiagen Plasmid Plus Midi kit.

To introduce the barcode construct, 5 µg of purified plasmid was digested with 250 units of ApaI restriction enzyme (NEB) for 1 h at 37 °C. Then, 1.5 µl QuickCIP (NEB) was added, and the reaction was incubated at 37 °C for a further 30 min. The reaction was then run on an agarose gel, and the digested fragment was isolated and column purified. The barcode (ApaI_barcode, Supplementary Table 4) was designed to have 20 degenerate nucleotides interspersed with constant AT dinucleotides: [5×N]AT[5×N]AT[5×N]AT[5×N]. It is flanked by a sequence complementary to the sequence flanking the ApaI cut site in the attB-HA-V2R plasmid and was ordered as a single-stranded oligonucleotide from IDT. ApaI_barcode was made double stranded and amplified by primers amp_ApaI_barcode[F/R]. Then, the barcode was introduced into the plasmid via Gibson assembly using 60 ng of vector and 3 ng of barcode and incubated at 50 °C for 20 min. The reaction was then diluted 1:12 with water; 1.5 µl of this was electroporated into 25 µl of 10β high-efficiency electrocompetent *E. coli* cells. Recovery was done in 2 ml of SOC at 37 °C for 45 min, then 0.5% was plated onto LB-agar plate with ampicillin; the other 99.5% was split between four flasks of 15 ml LB-amp liquid culture. In the morning, the number of clones on the plate was used to estimate that each flask should contain ~25,000 transformants. Two flasks were discarded, and the other two were combined to arrive at an estimated library diversity of 50,000. These were further incubated another 12 h in 100 ml LB and finally purified using the Qiagen Plasmid Plus Maxi kit.

### Long read sequencing to associate variants with barcodes

To liberate the sequence of interest from the rest of the plasmid (that is the barcode and V2R coding sequence), 10 µg of plasmid was digested with 15 units of KpnI-HF and 6 units of FseI (NEB), and the fragment was then run on an agarose gel. The digested fragment was isolated and column purified.

SMRTbell libraries were prepared using the SMRTbell Express Template Prep Kit 2.0 (Pacific Biosciences (PacBio)). The V2R mutagenesis libraries were multiplexed with other libraries and ran across two SMRT Cells on the Sequel IIe instrument (PacBio). The PacBio sequencing data were analyzed with alignparse[80]. Reads were quality filtered by removing any read with an estimated error rate above $1 \times 10^{-4}$ for the V2R coding sequence, or above $1 \times 10^{-3}$ for the barcode. Then, consensus sequences were called using the alignparse.consensus.simple_mutconsensus method with default parameter settings. For further processing, only barcodes with variant call support of ≥2 were retained, alongside those with variant call support of 1, provided the variant exhibited multiple nucleotide changes compared with the wild type.

### Landing-pad cell line recombination

HEK293T LLP-iCasp9-Blast Clone 12 from Matreyek et al.[39] (hereafter referred to as landing-pad cells) were used for library integration,

expression and screening. The cells were a gift from K. Matreyek at the University of Washington. Cells were cultured in DMEM supplemented with 10% tetracycline-free FBS. For recombination, 10 million landing pad cells were plated onto a T175 cell culture flask. The following day, 20 µg of the library was combined with 20 µg of pCAG-NLS-Bxb1 and transfected with Lipofectamine 3000 (Thermo Fisher Scientific), per the manufacturer's instructions. After 48 h, the meidum was removed and replaced with doxycycline-containing medium (2 µg ml⁻¹ Sigma-Aldrich). Twenty-four hours later, the medium was replaced again with medium containing doxycycline as well as rimiducid (10 nM, Selleckchem). Rimiducid causes cell death in unrecombined cells, and substantial cell death was apparent after 24 h. At this point, the medium was replaced with medium containing only doxycycline, not rimiducid. Cells were then grown out and passaged when approaching 95% confluency, always in the presence of doxycycline. The recombined cell population was expanded, then cryopreserved in many vials and used for all experiments.

### Cell sorting

Drug treatment was done 24 h before sorting. Tolvaptan (Selleckchem, catalog no. S2593) was dissolved to 10 mM in DMSO then added to cell culture medium, for a final concentration of 10 µM. To dissociate cells, they were first washed once with PBS, then incubated with Trypsin-EDTA (0.05%) for 4 min at room temperature. Then cells were washed off the plate with medium, then pelleted and resuspended in blocking buffer (1% bovine serum albumin in phosphate-buffered saline). Cells were counted and 30 million–50 million cells were transferred to a new tube. Blocking buffer was added to attain 15 million cells ml⁻¹. Then, cells were incubated on a rotating wheel at 4 °C for 30 min. Following this, HA-Tag (6E2) monoclonal antibody Alexa Fluor 647 conjugate (no. 3444, Cell Signaling Technologies) was added to a final concentration of 1:100, and cells were again incubated on a rotating wheel at 4 °C for 60 min. At this point, cells were pelleted and the supernatant was removed, then resuspended in 5 ml blocking buffer with propidium iodide (1 µg ml⁻¹).

Cells were sorted on a BD FACSaria II and analyzed with onboard FACSDiva software or post-sorting with FlowJo. Cells were first filtered by forward scattering area and side scattering area, then single cells were isolated with forward scattering width and height. BFP-positive cells were filtered as unrecombined landing-pad cells, and propidium-iodide-positive cells were filtered as dead cells. Then, the remaining population of cells was sorted into four bins, on the basis of Alexa Fluor 647 signal intensity, that were designed to result in a similar number of cells in each bin (Supplementary Fig. 1g). For each replicate, about 10 million cells were collected in total. After sorting, cells were pelleted and frozen at −80 °C, and processed later.

### Sequencing library preparation

DNA was isolated from cell pellets using the DNeasy Blood & Tissue Kit (Qiagen) and eluted with 150 µl buffer EB. For each sample, 128 µl sample was amplified in 400 µl polymerase chain reactions (PCRs) across 8 PCR tubes using Q5 High-Fidelity DNA Polymerase (NEB). Primers contained partial Illumina adapters, variable degenerate bases to promote complexity on the flowcell, and sequence complementary to the barcode flanks to amplify the barcode (ApaI_barcode_seq_[F/R]_[3-5] N, Supplementary Table 4). The PCR program was 98 °C for 30 s; then 25 cycles of 98 °C for 15 s, 64 °C for 30 s, 72 °C for 30 s; then 72 °C for 30 s followed by 8 °C thereafter. The products from this reaction were cleaned up with NucleoSpin columns (Macherey-Nagel) and eluted in 50 µl water. Two microliters of the eluate were then amplified in a second PCR that appended the rest of the Illumina sequencing adapter as well as index sequences (PCR2_i[5/7]). The PCR program was 98 °C for 30 s; then 5 cycles of 98 °C for 15 s, 64 °C for 30 s, 72 °C for 30 s; then 72 °C for 30 s, followed by 8 °C thereafter. The products were

run on an agarose gel and the intended product isolated and column purified. Libraries were sequenced on an Illumina NovaSeq 6000 with 2×50 paired end reads.

### Sequencing data analysis and calculation of surface expression scores

Paired-end sequencing reads in fastq format were first merged with vsearch[81], then adapters were trimmed with cutadapt[82]. The remaining sequence corresponds to the barcode, which was compared with the full list of barcodes identified from the PacBio variant-barcode association and tallied. Then, for each variant, the frequency of that variant in each bin is calculated. This frequency is compared with the original number of cells sorted into each bin to estimate the number of cells of that genotype in each bin. Then, all barcode-variants that code for the same amino acid change are combined, and the $\log_{10}$-transformed geometric mean fluorescence (fluor) value of all cells sorted into each bin is combined with the estimated cell counts to arrive at the raw surface expression estimate:

$$\frac{\begin{array}{c}((\text{bin-1 no.cells} \times \text{bin-1fluor}) + (\text{bin-2 no.cells} \times \text{bin-2fluor}) \\ + (\text{bin-3 no.cells} \times \text{bin-3fluor}) + (\text{bin-4 no.cells} \times \text{bin-4fluor}))\end{array}}{\text{bin-1 no.cells} + \text{bin-2 no.cells} + \text{bin-3 no.cells} + \text{bin-4 no.cells}}$$

Then, the surface expression scores were normalized such that the wild-type genotype was 1 and the median of known loss-of-function variants (premature stop codons before the 300th residue) was 0. Variants were retained and considered high confidence if the estimated number of cells sorted for that genotype was ≥50.

For the control scores, four replicates were combined- these were the controls for the rescue experiments. Namely, one condition with normal culture conditions (37 °C condition), two conditions with 0.1% DMSO (DMSOA and DMSOB, controlling for tolvaptan) and one condition with 1% DMSO. Because these replicates were well correlated with each other, they were all used to derive the control estimates. Data analysis was performed in jupyter lab environments using Python and the following packages: matplotlib, numpy, seaborn, pandas, scipy, sklearn and statsmodels.

### Modeling rescue and identifying outliers

LOWESS curves were fit to the controls versus rescue data with the Python package statsmodels.nonparametric.smoothers_lowess[83] with these parameter values: frac=0.3, it=3, delta=0.0. After fitting the LOWESS curve, residuals were calculated for all variants as the distance in the $y$ axis between the curve and the variant. Then, for each position, a two-sided Mann–Whitney $U$ test was conducted, comparing the residuals of all variants at that position with control surface expression below 0.85 with all other variants with control surface expression below 0.85. This threshold was intended to include only variants that actually had the potential to be rescued. To control the FDR, we then used statsmodels.stats.multitest.multipletests, method = "fdr_bh", alpha=0.1.

### Clinical variant curation

The gnomAD data was from version v2.1.1 with controls only, and was obtained on 8 February 2024. Clinvar variants were obtained on 9 February 2024 and filtered for missense variants. HGMD variants were obtained on 8 February 2024 and filtered for missense variants.

### Computational predictors

ESM1b was installed and ran as described (https://github.com/ntra-noslab/esm-variants) using the esm_score_missense_mutations.py function. AlphaMissense, EVE and RaSP scores were available as precomputed scores that we downloaded from the respective servers. ThermoMPNN scores were computed using the Google Colab server.

### Reporting summary

Further information on research design is available in the Nature Portfolio Reporting Summary linked to this article.

### Data availability

Files needed to reproduce analyses can be found at Zenodo (https://zenodo.org/records/14216036)[84]. Raw sequencing reads can be found at Sequence Read Archive (accession number PRJNA1190688). Clinical annotations taken from ClinVar (https://www.ncbi.nlm.nih.gov/clinvar/), Human Gene Mutation Database (https://www.hgmd.cf.ac.uk/ac/index.php) and gnomAD (https://gnomad.broadinstitute.org/). Data and materials can be obtained from the corresponding author upon request. Source data are provided with this paper.

### Code availability

Custom code to reproduce analyses can be found at github (https://github.com/lehner-lab/V2R_surfexp_rescue).

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

### Acknowledgements

This work was funded by a European Research Council Advanced Grant (883742), Wellcome (220540/Z/20/A), the Spanish Ministry of Science and Innovation (LCF/PR/HR21/52410004, EMBL Partnership, Severo Ochoa Centre of Excellence), the Bettencourt Schueller Foundation, the AXA Research Fund, Agencia de Gestio d'Ajuts Universitaris i de Recerca (AGAUR, 2017 SGR 1322) and the CERCA Program/Generalitat de Catalunya. T.L.M. was funded by an EMBO fellowship (ALTF 113-2021). We thank all members of the Lehner Lab and J. Selent and T. Stepniewski for helpful discussions. We thank the CRG/UPF Flow Cytometry Unit for assistance with the sorting experiments.

### Author contributions

The project was conceived by T.L.M. and B.L. Experiments and analyses were performed by T.L.M. Figures were prepared by T.L.M. The manuscript was written by T.L.M. and B.L.

### Competing interests

B.L. is a founder and shareholder of ALLOX. T.L.M. declares no competing interests.

### Additional information

**Extended data** is available for this paper at https://doi.org/10.1038/s41594-025-01659-6.

**Correspondence and requests for materials** should be addressed to
Ben Lehner.

**Peer review information** *Nature Structural & Molecular Biology*
thanks Willow Coyote-Maestas, Kenneth Matreyek and Jonathan
Schlebach for their contribution to the peer review of this work.

Primary Handling Editor: Katarzyna Ciazynska, in collaboration
with the *Nature Structural & Molecular Biology* team. Peer reviewer
reports are available.

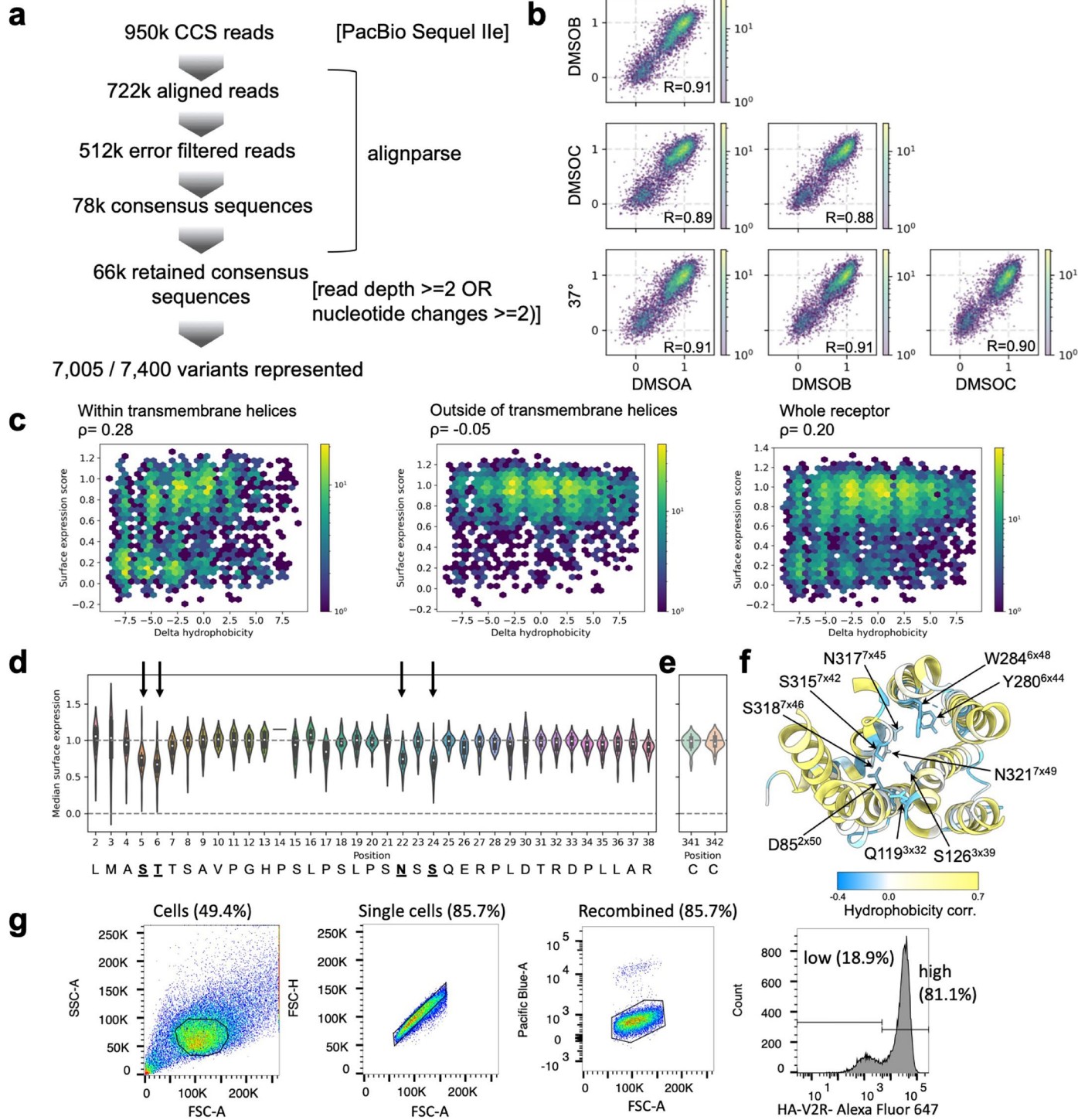

**Extended Data Fig. 1 | Extended analyses of primary V2R surface expression data. a** Overview of data processing for barcode-variant association with PacBio long read sequencing. **b** Replicate correlations for the four control conditions. 37° was culture media without DMSO, DMSOA and DMSOB were culture media with 0.1% DMSO, DMSOC was culture media with 1% DMSO. Because the conditions are all well correlated, they were all considered control and combined for the control estimates. P < 1×10^100 in all cases. **c** Hexbin plots comparing the change in hydrophobicity with surface expression score, separated by positions that are in transmembrane domains, outside of transmembrane domains, or the whole receptor considered together. **d** Violin plot of variant effects in the N-terminus of V2R. Highlighted with arrows are the putative O- and N-glycosylation sites (5 and 6, 22 and 24, respectively). **e** Variant effects at the sites of palmitoylation. **f** Topdown view of V2R, with residues colored by their preference for hydrophobicity. Highlighted with arrows are residues in the core of the receptor where hydrophilic amino acids are preferred. **g** Flow cytometry gating strategy.

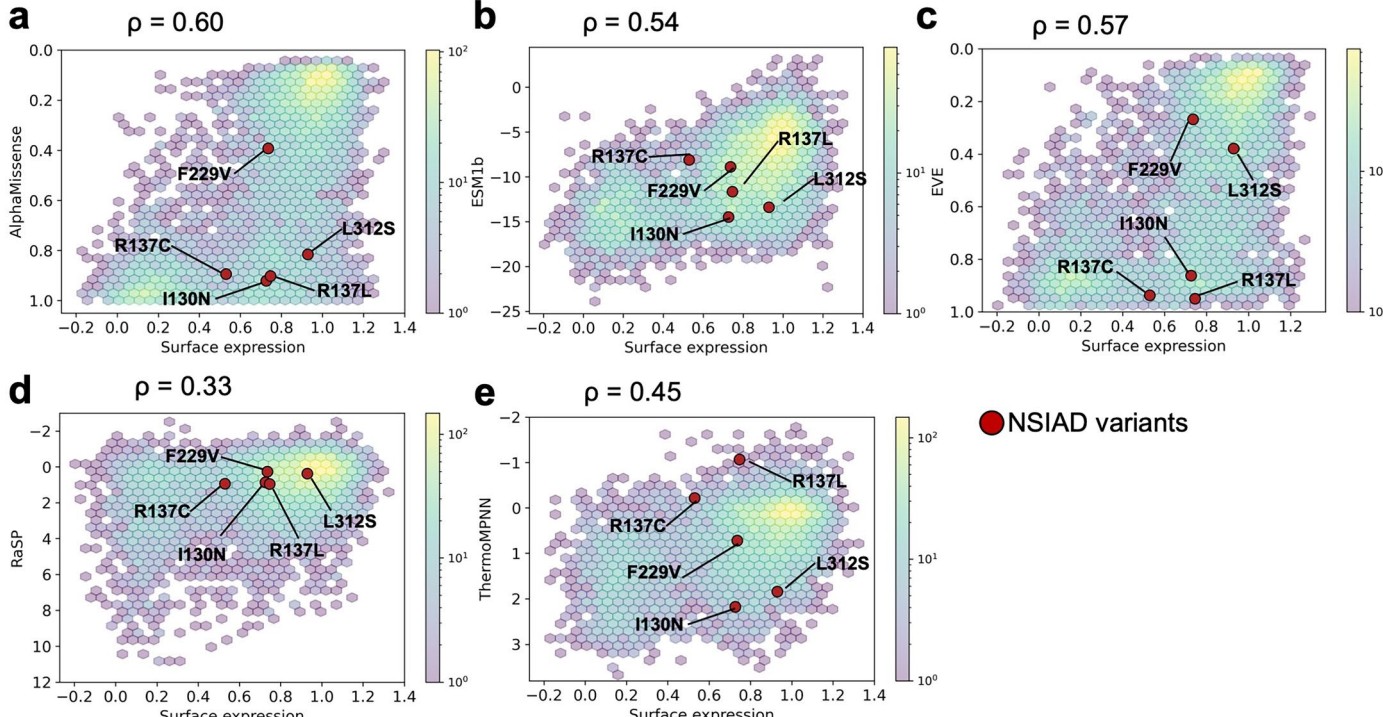

**Extended Data Fig. 2 | Comparison of computational VEPs with empirical surface expression scores, highlighting NSIAD scores. a** Hexbin plot comparing surface expression with AlphaMissense scores for all missense variants. NSIAD scores are highlighted. **b** Hexbin plot comparing surface expression with ESM1b scores for all missense variants. NSIAD scores are highlighted. **c** Hexbin plot comparing surface expression with EVE scores for all missense variants. NSIAD scores are highlighted. **d** Hexbin plot comparing surface expression with RaSP scores for all missense variants. NSIAD scores are highlighted. **e** Hexbin plot comparing surface expression with ThermoMPNN scores for all missense variants. NSIAD scores are highlighted.

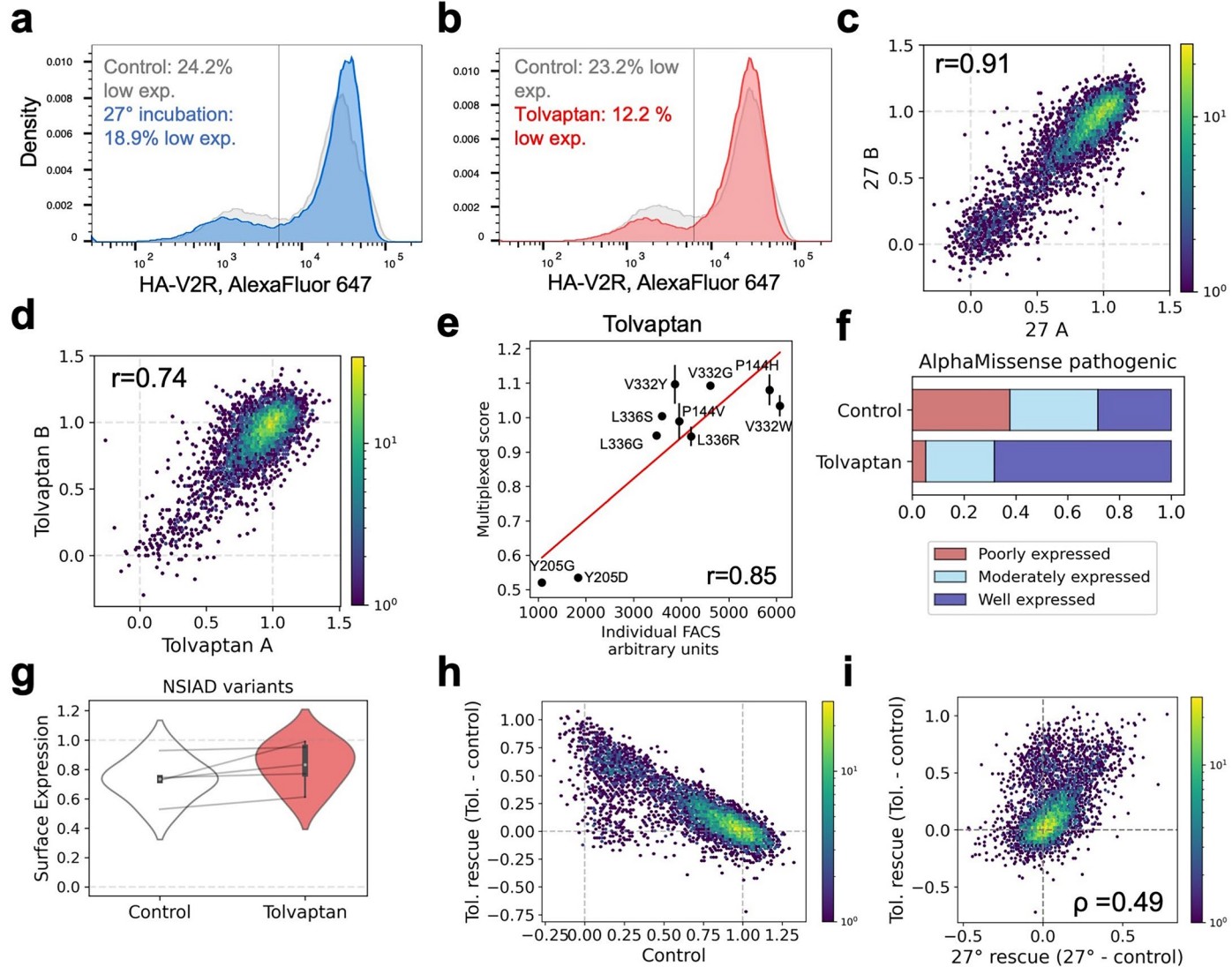

**Extended Data Fig. 3 | Extended analysis of rescue experiments. a** FACS data for V2R library in the control versus reduced temperature (27°) condition. **b** FACS data for V2R library in the control versus Tolvaptan condition. **c** Replicate correlations for 27° experiment. p < 1×10^100. **d** Replicate correlations for Tolvaptan experiment. p < 1×10^100. **e** Comparison of ten variants measured in multiplex or individually, in the presence of Tolvaptan. **f** Percent of AlphaMissense predicted pathogenic variants that are poorly, moderately, or well expressed in control and Tolvaptan conditions. **g** Violinplot comparing the surface expression scores of NSIAD variants in the control compared with the Tolvaptan condition. **h** Control surface expression compared with Tolvaptan rescue magnitude. **i** Reduced temperature rescue compared with Tolvaptan rescue magnitude. p < 1×10^100.

# Reporting Summary

## Statistics

For all statistical analyses, confirm that the following items are present in the figure legend, table legend, main text, or Methods section.

| n/a | Confirmed | |
|---|---|---|
| ☐ | ☒ | The exact sample size (*n*) for each experimental group/condition, given as a discrete number and unit of measurement |
| ☐ | ☒ | A statement on whether measurements were taken from distinct samples or whether the same sample was measured repeatedly |
| ☐ | ☒ | The statistical test(s) used AND whether they are one- or two-sided <br> *Only common tests should be described solely by name; describe more complex techniques in the Methods section.* |
| ☒ | ☐ | A description of all covariates tested |
| ☐ | ☒ | A description of any assumptions or corrections, such as tests of normality and adjustment for multiple comparisons |
| ☐ | ☒ | A full description of the statistical parameters including central tendency (e.g. means) or other basic estimates (e.g. regression coefficient) AND variation (e.g. standard deviation) or associated estimates of uncertainty (e.g. confidence intervals) |
| ☐ | ☒ | For null hypothesis testing, the test statistic (e.g. *F*, *t*, *r*) with confidence intervals, effect sizes, degrees of freedom and *P* value noted <br> *Give P values as exact values whenever suitable.* |
| ☒ | ☐ | For Bayesian analysis, information on the choice of priors and Markov chain Monte Carlo settings |
| ☒ | ☐ | For hierarchical and complex designs, identification of the appropriate level for tests and full reporting of outcomes |
| ☐ | ☒ | Estimates of effect sizes (e.g. Cohen's *d*, Pearson's *r*), indicating how they were calculated |

*Our web collection on statistics for biologists contains articles on many of the points above.*

## Software and code

Policy information about availability of computer code

Data collection: FACSDiva version 8.0.2 software used onboard the FACSaria instrument.

Data analysis:
ChimeraX version 1.6
FlowJo version 10.8.0
python version 3.8.3
matplotlib version 3.5.2
numpy version 1.21.5
seaborn version 0.11.2
pandas version 1.4.2
scipy version 1.8.0
sklearn version 1.0.2
statsmodels version 0.13.2
cutadapt version 2.4
vsearch version 2.22.1

Custom code to reproduce analyses can be found at: https://github.com/lehner-lab/V2R_surfexp_rescue

For manuscripts utilizing custom algorithms or software that are central to the research but not yet described in published literature, software must be made available to editors and reviewers. We strongly encourage code deposition in a community repository (e.g. GitHub). See the Nature Portfolio guidelines for submitting code & software for further information.

## Data

Policy information about availability of data

All manuscripts must include a data availability statement. This statement should provide the following information, where applicable:

- Accession codes, unique identifiers, or web links for publicly available datasets
- A description of any restrictions on data availability
- For clinical datasets or third party data, please ensure that the statement adheres to our policy

> Files needed to reproduce analyses can be found at zenodo (https://zenodo.org/records/14216036). Raw sequencing reads can be found at Sequence Read Archive (accession number PRJNA1190688). Clinical annotations taken from ClinVar (https://www.ncbi.nlm.nih.gov/clinvar/), Human Gene Mutation Database (https://www.hgmd.cf.ac.uk/ac/index.php) and gnomAD (https://gnomad.broadinstitute.org/).

## Research involving human participants, their data, or biological material

Policy information about studies with human participants or human data. See also policy information about sex, gender (identity/presentation), and sexual orientation and race, ethnicity and racism.

| | |
|---|---|
| Reporting on sex and gender | Our study does not use any sex or gender aspect. |
| Reporting on race, ethnicity, or other socially relevant groupings | Our study does not use any race, ethnicity, or other socially relevant groupings. |
| Population characteristics | This study does not use human research participants. |
| Recruitment | There were no participants in this study. |
| Ethics oversight | We did not have approval as there were no human participants. |

Note that full information on the approval of the study protocol must also be provided in the manuscript.

# Field-specific reporting

Please select the one below that is the best fit for your research. If you are not sure, read the appropriate sections before making your selection.

☒ Life sciences   ☐ Behavioural & social sciences   ☐ Ecological, evolutionary & environmental sciences

For a reference copy of the document with all sections, see nature.com/documents/nr-reporting-summary-flat.pdf

# Life sciences study design

All studies must disclose on these points even when the disclosure is negative.

| | |
|---|---|
| Sample size | We sought to include all single amino acid substitutions of V2R in this study. |
| Data exclusions | Variant measurements in which the variant was estimated to be present in less than 50 cells was excluded. |
| Replication | All experimental conditions were replicated, and results were in agreement. For the control condition, four independent experiments were performed. For the temperature and Tolvaptan rescue conditions, two replicates were performed for each condition. |
| Randomization | This is not relevant- the same library of variants was used in all conditions. |
| Blinding | Blinding is not relevant to this experiment. All variants in all conditions are analyzed with the exact same procedure. |

# Reporting for specific materials, systems and methods

We require information from authors about some types of materials, experimental systems and methods used in many studies. Here, indicate whether each material, system or method listed is relevant to your study. If you are not sure if a list item applies to your research, read the appropriate section before selecting a response.

## Materials & experimental systems

| n/a | Involved in the study |
|---|---|
| ☐ | ☒ Antibodies |
| ☐ | ☒ Eukaryotic cell lines |
| ☒ | ☐ Palaeontology and archaeology |
| ☒ | ☐ Animals and other organisms |
| ☒ | ☐ Clinical data |
| ☒ | ☐ Dual use research of concern |
| ☒ | ☐ Plants |

## Methods

| n/a | Involved in the study |
|---|---|
| ☒ | ☐ ChIP-seq |
| ☐ | ☒ Flow cytometry |
| ☒ | ☐ MRI-based neuroimaging |

## Antibodies

| | |
|---|---|
| Antibodies used | HA-Tag (6E2) monoclonal antibody Alexa Fluor 647 Conjugate (#3444, Cell Signaling Technologies) |
| Validation | This is a widely used antibody for a non-species specific epitope. Validation on supplier's website: "Flow cytometric analysis of COS cells, untransfected (blue) or transfected with HA-tagged DLL1 (green), using HA-Tag (6E2) Mouse mAb (Alexa Fluor 647 Conjugate) |

## Eukaryotic cell lines

Policy information about <u>cell lines and Sex and Gender in Research</u>

| | |
|---|---|
| Cell line source(s) | The cell line is HEK293T LLP-iCasp9-Blast Clone 12 described in this publication: https://academic.oup.com/nar/article/48/1/e1/5587635. The cells were provided as a gift by Kenny Matreyek from the University of Washington |
| Authentication | The cell line was not authenticated. |
| Mycoplasma contamination | The cell line tested negative for mycoplasma. |
| Commonly misidentified lines (See <u>ICLAC</u> register) | None. |

## Plants

| | |
|---|---|
| Seed stocks | We did not use seed stocks. |
| Novel plant genotypes | We did not use any plants. |
| Authentication | We did not use any plants. |

## Flow Cytometry

### Plots

Confirm that:

☒ The axis labels state the marker and fluorochrome used (e.g. CD4-FITC).

☒ The axis scales are clearly visible. Include numbers along axes only for bottom left plot of group (a 'group' is an analysis of identical markers).

☒ All plots are contour plots with outliers or pseudocolor plots.

☒ A numerical value for number of cells or percentage (with statistics) is provided.

### Methodology

| | |
|---|---|
| Sample preparation | Drug treatment was done 24 hours prior to sorting. Tolvaptan (Selleckchem, catalog number S2593) was dissolved to 10 mM in DMSO then added to cell culture media for a final concentration of 10 μM. To dissociate cells, they were first washed once with PBS, then were incubated with Trypsin-EDTA (0.05%) for 4 minutes at room temperature. Then cells were washed off the plate with media, then pelleted and resuspended in blocking buffer (1% bovine serum albumin in phosphate buffered saline). Cells were counted and 30-50M cells were transferred to a new tube. Blocking buffer was added to attain 15M cells/mL. Then, cells were incubated on a rotating wheel at 4° for 30 minutes. Following this, HA-Tag (6E2) monoclonal antibody |

Alexa Fluor 647 Conjugate (#3444, Cell Signaling Technologies) was added to a final concentration of 1:100, then cells were again incubated on a rotating wheel at 4° for an additional 60 minutes. At this point, cells were pelleted and supernatant removed, then resuspended in 5 mL of blocking buffer with propidium iodide (1 μg/mL).

| Instrument | Cells were sorted on a BD FACSaria II. |
| --- | --- |
| Software | onboard FACSDiva software was used to analyze data. Flowjo was used to visualize data. |
| Cell population abundance | After the gating for recombined and alive cells, all cells were sorted and collected. |
| Gating strategy | Cells were first filtered by forward scattering area and side scattering area, then single cells were isolated with forward scattering width and height. BFP positive cells were filtered as unrecombined landing pad cells, and propidium iodide-positive cells were filtered as dead cells. Then, the remaining population of cells was sorted into four bins, based on Alexa Fluor 647 signal intensity, that were designed to result in a similar number of cells in each bin. |

☒ Tick this box to confirm that a figure exemplifying the gating strategy is provided in the Supplementary Information.

