## [Peer Review File · Nature Structural & Molecular Biology]

A small molecule stabilizer rescues the surface expression of nearly all missense variants in a GPCR

Corresponding Author: Professor Ben Lehner

Version 0:

Decision Letter:

11th Mar 2025

Dear Dr. Lehner,

Thank you again for submitting your manuscript "A pharmacological chaperone stabilizer rescues the expression of the vast majority of pathogenic variants in a G protein-coupled receptor". I apologize for the delay in responding, which resulted from the difficulty in obtaining suitable referee reports. Nevertheless, we now have comments (below) from the 3 reviewers who evaluated your paper. In light of those reports, we remain interested in your study and would like to see your response to the comments of the referees, in the form of a revised manuscript.

You will see that while reviewers appreciate the results, they raise several concerns which will need to be addressed in a revision. Specifically, you will see that the reviewers agree that some claims need to be toned down – especially where claims are not supported by orthogonal approaches, as well as claims of novelty. We would ask that you address these comments textually, and put the findings in the context of the wider literature brought up in review. You will also see that R#2 in particular would like to see the final model revised, which provides a more appropriate view of current knowledge of membrane protein biogenesis. Furthermore, we agree with reviewer #2 that experimental investigation into the separate effects of PCs and temperature will strengthen the manuscript. We also agree with reviewers #1 and #2 that characterising rescue of function, in addition to surface expression of V2R, will be of interest, and should be pursued experimentally, if feasible.

Please be sure to address/respond to all concerns of the referees in full in a point-by-point response and highlight all changes in the revised manuscript text file. If you have comments that are intended for editors only, please include those in a separate cover letter.

We expect to see your revised manuscript within 6 weeks. If you cannot send it within this time, please contact us to discuss an extension; we would still consider your revision, provided that no similar work has been accepted for publication at NSMB or published elsewhere.

Reporting Summary:

When submitting the revised version of your manuscript, please pay close attention to our <https://www.nature.com/nature-portfolio/editorial-policies/image-integrity> Digital Image Integrity Guidelines and to the following points below:

EXTENDED DATA FIGURES

Please note that all key data shown in the main figures as cropped gels or blots should be presented in uncropped form, with molecular weight markers. These data can be aggregated into a single supplementary figure item. While these data can be displayed in a relatively informal style, they must refer back to the relevant figures. These data should be submitted with the final revision, as source data, prior to acceptance, but you may want to start putting it together at this point.

Data availability: this journal strongly supports public availability of data. All data used in accepted papers should be available via a public data repository, or alternatively, as Supplementary Information. If data can only be shared on request, please explain why in your Data Availability Statement, and also in the correspondence with your editor. Please note that for some data types, deposition in a public repository is mandatory - more information on our data deposition policies and available repositories can be found below:

<https://www.nature.com/nature-research/editorial-policies/reporting-standards#availability-of-data>

Link Redacted

Sincerely,

Katarzyna Ciazynska, PhD
(she/her)

Senior Editor
Nature Structural & Molecular Biology
<https://orcid.org/0000-0002-9899-2428>

Referee expertise:

Referee #1: DMS, molecular genetics

Referee #2: DMS, membrane proteins

Referee #3: protein folding

Reviewers' Comments:

Reviewer #1 (Remarks to the Author):

In this manuscript, the authors perform and analyze the results of a cell-surface trafficking assay for missense variants of the vasopressin 2 receptor (V2R) protein in cultured human cells. The authors first use a well-established Sort-Seq approach to sort cells each expressing individual members from a large library of V2R single amino acid variants based on fluorescent antibody binding, indicative of the amount of variant V2R protein that made it to the cell surface. The authors repeat the experiment at reduced temperatures and pre-incubate with the V2R antagonist, tolvaptan. The authors identify that the vast majority of V2R loss-of-function variants are caused by presumed protein destabilization, and that cell-surface expression of these proteins can be rescued by tolvaptan, with clear implications for pharmacological chaperones (PCs) as a generalizable means to rescue destabilizing variants in membrane proteins. The main feature of this paper is that the authors provide a significant advancement in potential clinical considerations for PCs. While the experiments and the ensuing analyses appear well performed, the manuscript suffers from exaggerative language that overstates the potential impacts of the results, which need to be revised prior to publication.

Major comments:

1) It's unclear what the generality of the PC phenomenon is across protein targets and putative PCs. Only in the discussion do the authors introduce similar, albeit smaller-scale studies performed with Rhodopsin (another GPCR!) and CFTR, that seem to support the alternative hypothesis of PCs being far more selective in the proportion of destabilizing missense variants it rescues. These studies should be mentioned much earlier in the manuscript, so that the potential universality of the results is better contextualized and understood by the reader. These differences are particularly stark, as all studies (including a recently published manuscript in PLoS Biology on Rhodopsin cell-surface trafficking rescue by PCs; PMID: 39808594) seem to utilize very similar transgenic overexpression systems, albeit with different gene targets and candidate PCs. Do the authors have any ideas as to why they observed such large-scale rescue of variant trafficking for V2R with tolvaptan, while the other authors observed more region-specific rescue of variants with their protein and drug targets? Could it be the location of small molecule binding? Or could it be related to the amount of stabilization conferred by small molecule binding, which will drastically differ depending on the protein variant target and small molecule in question? This information would seem to help the reader understand whether the effect observed with Tolvaptan and V2R cell surface rescue is expected to be an unusual outlier case, or whether it's likely to be more commonplace across contexts.

2) What is the relationship between surface expression and function? This method measures surface expression, but the authors don't directly show any evidence showing that rescued trafficking correlates perfectly with rescued function. This is particularly important for the Tolvaptan PC experiments. Tolvaptan is described in the manuscript as a "selective antagonist", with a hypothesized binding site overlapping the binding site for the hormone arginine vasopressin (AVP). A quick literature search seems to suggest that it indeed counteracts the signaling pathway normally initiated by AVP binding, thus pharmacologically conferring a loss of function to V2R. Thus, it would seem that while tolvaptan may bind to V2R and help rescue its cell surface localization, that its constant presence may also interfere with the receptor's ability to bind its endogenous ligand and transmit signal, thus not functionally rescuing the variants. It would seem that NDI-associated variants, which are currently largely loss-of-function through destabilization and mistrafficking, may simply become properly trafficked but still non-functional variants in the presence of tolvaptan. Experimental evidence demonstrating functional rescue, or citations demonstrating the functional rescue of endogenous AVP signaling upon co-incubation with Tolvaptan, seem necessary to support any potential functional implications of rescue. Relatedly, the potential for tolvaptan to inhibit WT copies while rescuing the cell surface trafficking of destabilized variants seem to potentially impact prospects for using this drug in NDI affected females.

3) Disconnect and inconsistency in strength of conclusions made in the manuscript: eg. in the abstract "Strikingly, treatment with a PC rescues the expression of nearly all destabilized variants, with non-rescued variants identifying the drug's binding site" while in the results section "The few sites where variants are consistently not rescued inform on the likely binding site of Tolvaptan as well as other functional sites of the receptor". The latter statement seems more appropriate, unless experimental evidence showing drug binding at that location is provided. With nearly the entirety of the work and analysis resulting from data from a singular assay (repeated in different conditions), I do think it is generally more prudent to leave many of these interpretations outside of cell-surface localization as possibilities or likely implications, rather than definitive statements.

Minor comments:

- 4) Yellow outline visualization scheme in Fig 5 is really hard to see.
- 5) Line 326-328: confusing grammar; would suggest rephrasing for clarity.

Reviewer #2 (Remarks to the Author):

Summary:

Mighell & Lehner conduct a series of deep mutational scanning experiments on the Vasopressin 2 Receptor (V2R) where they investigate mutational effects on receptor surface expression in the presence of reduced temperature or a pharmacochaperone (PC). They show that both reduced growth temperature (27C) and PC (tolvaptan) treatment can improve expression of normally poorly-expressing variants. Results are interpreted in the context of known pathogenic human variants, structural data, and a variety of thermodynamic and other variant effect predictors. Overall this work increases our understanding of sequence-to-expression determinants in GPCRs and the mechanism of temperature and PC chaperone effects. That said the manuscript would be greatly improved through a major revision by putting the work in context of past work in the membrane protein biogenesis field and reducing the generalizations made based solely on this experiment on a single receptor and single pharmacochaperone vs temperature.

Major points

Strengths

The authors embark on the first of its kind experiment measuring the how temperature and a small molecule chaperone rescue all mutations in a membrane protein, V2R. As small molecule chaperones have transformed the treatment of cystic fibrosis and many in industry are embarking on similar campaigns in other membrane proteins, understanding the mechanisms of how chaperones rescue mutations will be broadly useful. Similarly, that temperature rescue was done here can begin to answer questions on the mysterious nature of temperature rescue.

Membrane protein biogenesis is an important and conventionally intractable area of study. The authors add a critical new mutational scanning dataset that can help understand how mutations break membrane proteins and in turn how residues contribute to surface expression.

Opportunities for improvement

In the first line of the abstract authors say reduced protein stability is the most common effect of mutations. I think while it is true that protein abundance is the most common effect of mutations, stability is a specific property referring to the life-time of the protein at least in the context of membrane proteins. We believe this confusion is likely coming because in the protein folding space - fold stability is used as a catch-all term.. However, for membrane proteins where mutations can alter biogenesis (which is a catch-all term to mean the formation of protein) vs a protein's stability - this is an inaccurate statement. Probably better just to use the word 'abundance'. For instance, if the authors are curious to know where these classifications come from we recommend they read the CF variant classification literature where biogenesis and stability are distinct mechanisms and most of the pharmacochaperones here are acting upon biogenesis likely but without a distinct measure that separates these it's hard to differentiate the effects. In general, I would recommend the authors change the language throughout the manuscript to represent this nuance because while pedantic - this paper will be read by the membrane protein biogenesis/stability/folding field and these terms must be referred to specifically. For instance the citations the authors use were not measured in the context of differentiating the formation vs the degradation of proteins - so that statement is not backed up by the literature.

Currently, the figures seem to have many many panels and be full page figures. For instance the first figure has many comparisons that may not need to be in the main figure such as the membrane insertion indices or the hydrophobicity scores. Similarly, figure 2 has many different comparisons of pathogenic predictors - these sorts of studies have been done previously and I did not learn much from these panels. Figure 3, similarly seems to take up a tremendous amount of space with concepts of how a LOWESS fit works. In general we recommend the authors work on focusing the figures and panels on the observations and conclusions they most want to highlight in the manuscript.

The most interesting and novel part of this study is measuring how reducing the temperature and adding pharmacochaperones for all possible mutations effect expression of V2R. However, the authors don't compare between these these two conditions. These are really different mechanisms of rescue and we think it would be worth spending more space directly comparing the two and exploring how they may be different. Are the positions rescued more or less different between the two conditions? In general, temperature based rescue of expression is a bit mysterious as it's unlikely to be due to thermodynamic stability and could more be due to slowing down the processes of biogenesis and therefore increasing the fidelity of surface expression. In contrast the binding of a pharmacochaperone is likely to stabilize specific states in

biogenesis. If the residues that rescue both are the same but just a difference in magnitude that is worth explicitly discussing. If not, it also very interesting and worth highlighting.

In the discussion, the authors present the work as if it is contradictory to the field and prior results. While this study is the first study of how a single pharmacochaperone rescues all single mutations in V2R, there have been many previous studies on the mechanisms of other pharmacochaperones in other membrane proteins. The authors discuss the work as if pharmacochaperones (based on their work) are a novel approach for curing disease, but this has previously been established. The authors seem to argue that most mutants are rescued by a single chaperone implies that prior results on CFTR in which different pharmacochaperones rescue different subsets of mutations is wrong. While this is an interesting example, the authors have not sufficiently tested this hypothesis and do not have other compounds to compare to be able to be convincing. It is perhaps unsurprising that different protein architectures (GPCR vs transporter) have different results. We encourage the authors to consider not over-weighting the results on this single protein with a single compound to override results in completely different proteins that include structural characterization and more biophysical measurements.

Figure 6 gives a misleadingly simple thermodynamic model of the 'folding' of V2R. Membrane protein biogenesis is complex and the experiments done here do not give sufficient information to propose such a model. For instance, the authors are measuring surface expression - while it is likely most of the effects of a mutation are due to biogenesis or stability - the authors do not necessarily know which variants may just be affecting surface expression. We recommend removing this completely and ending on figure 5.

Minor Points:

Line 102: The authors state they test the hypothesis that pharmacochaperones can stabilize mutations across a 'complex protein'. While this study is novel in that it measures the effects of all mutations, Jon Schleich's group has previously tested this hypothesis in several hundred mutations that are distributed across CFTR. More recently the cystic fibrosis foundation tested ~700 mutations in laborious assays that confirm this result ([https://www.cysticfibrosisjournal.com/article/S1569-1993\(24\)00021-3/fulltext](https://www.cysticfibrosisjournal.com/article/S1569-1993(24)00021-3/fulltext)). We recommend the authors change language that don't claim to have discovered this for the first time. Further, it's unclear what a 'complex protein' is here - perhaps be a bit more specific.

Line 110: Similarly, here the authors say their results provide proof that a pharmacochaperone can rescue mutations across the receptor - this is well-known from the same CF literature as the prior comment.

Line 149: We found the normalization of using synonymous and nonsense variants confusing here but this is probably my reading. Is it that all nonsense and synonymous variants are set at 1 or 0 or is it that the distribution is set? Reading the corresponding section of the methods didn't really clear up my confusion. More detail would be useful. Has this approach been used previously? It seems there is no literature cited here but many others have done FACS based screens - it would be useful to understand whether this is what others use for similar experiments.

Line 169: The authors state that previous work described TM3 plays a critical role in receptor stability across all class A GPCRs. However, the papers they cite are computational models and have little experimental basis. There have been a number of DMS experiments done on GPCRs and extensive studies of biogenesis and stability in rhodopsin - in general it doesn't seem that this model has been proven true across these datasets. Line 174: The authors compare mutational effects to membrane insertion indices qualitatively. In Figure 1F TM7 has a higher insertion Δ IG than TM3 but the authors state that TM3 has amongst the lowest expression and that maybe is because of the insertion index. I find this incongruency a bit confusing. Perhaps either directly compare the two quantitatively or don't include the membrane insertion indices and parsed TM scores in 1E altogether.

Lines 176-185: The authors compare the effect of mutations that reduce hydrophobicity. I think some explanation of why this analysis was done would be worthwhile here. Furthermore it is expected that residues facing the membrane vs those that don't will have different effects. Perhaps the authors should compare this.

Line 186 The authors compare the effects of mutations in glycosylation sites. The authors describe 'strong' mutational effects - what is a strong effect? Is this similar to mutations within membrane facing residues? In some ways the alternative explanation here is that any single glycosylation is not critical yet they combined together effect expression. Furthermore, we suggest caution in making these claims especially in the context of an over-expression system in which the assay may not be able to detect the effect of removing a single glycosylation site.

Line 200: The authors compare surface expression to pathogenic classifications. There are no statistical tests here - we suggest the authors add these. Related, figure 2A and 2B seem slightly redundant, perhaps consider condensing.

Figure 2d - The authors label VEP plus mechanistic screen. We feel calling this a mechanistic screen is a bit much as there is no readout for activity. This is evidenced by the statement in line 253 of 'presumed signaling defect'. We caution against the use of this term here.

Figure 2H-J captions are missing from the legend.

Line 281. The authors state that they are measuring thermodynamic rescue when they lower the temperature the cells are grown at prior to the screen. Temperature rescue is likely acting through many different mechanisms - some of which may be thermodynamic - others may be slowing down the process and increasing fidelity. Temperature rescue of CFTR variants has

a long history such as: <https://pubmed.ncbi.nlm.nih.gov/1380673/>. And is widely used in structural biology to increase the level of a membrane protein prior to purification. While the temperature rescue DMS is a beautiful experiment, we urge the authors to remove the word thermodynamic in the title and throughout the text and instead use something more descriptive such as 'temperature rescue'.

Figure 3. The authors spend much of the figure and the corresponding text describing the data processing but not the distributions of the temperature rescued variants. For instance, panel E could be condensed to a single example plot. Overall, we are left with many questions, such as where are these variants in the protein? Are there any trends with physicochemistry, TM#, etc? W296 is called out as having the "most significantly different" set of residuals but not really discussed, what are some of the other top hits or interesting findings?

Line 314 (related to the previous comment). The authors mention that mutations in rescued are rescue more (6 residues) or less (4) than would be expected from their expression level. This is a bit confusing. What would be expected? Perhaps it would be worth unpacking this a bit more. Further, which ones are more than would be expected and which and which ones are less? Where are these? The authors mention glycosylation sites. Are some glycosylation sites being rescued more while others less? What would this mean? There is a more detailed look at this in supplemental figure 1, but perhaps it would be worth moving the key bits to the main text?

Supplementary figure 1F. Include the mentioned scale bar from fig 1h.

Line 341 the authors mention that Tolvaptan has a stronger effect than temperature at rescuing cells and reference S Fig3b. While it looks like there is a slight effect it's hard to tell how strong this effect is. I would recommend switching the Plots of supplemental Figure 3a-b to being stacked vs overlaid and reporting numbers and percentages of cells in these populations. That would make it easier to judge this claim but also to see what the increased enrichment is.

Line 366 the authors discuss that there are more positions that have specific interactions w/ Tolvaptan than temperature (32 vs 10). Are all of the 10 shared? If not, why? I think this is the coolest part of the paper and is worth discussing. This demonstrates the power of this systematic approach - to describe specific effects across ALL positions - and is what sets this paper apart from prior studies. Discussing this more would illustrate the strength of the approach more explicitly and could lead to meaningful insights for the membrane protein folding field.

Figure 4H-I, similar to the comment on the previous figure, S171 is called out in the figure but not really discussed in the text with this section, or even in the structural interpretation with figure 5. There is another position in the top left of the plot adjacent to 171, what is this? What about the other significant positions? A little bit of unpacking might be needed here.

Figure 5 and associated text. The authors should mention what the residue superscript means, the GPCRdb/BW numbering system is a bit of a niche thing for GPCRs and not always immediately understood by a broad audience.

Figure 5. We'd encourage the authors to find a new color instead of yellow to highlight residues, it is challenging to read/see this in both print and on a computer screen.

Line 390. The authors describe the effects of the highly conserved DRY motif. A brief sentence describing the motif and its role could be helpful for broad readability. There have been a number of GPCR DMS studies conducted including several with surface expression readouts. It may be worth specifically discussing whether mutational effects to this or other canonical GPCR motifs/microswitches generally alter expression.

Related to figure 5b3, the authors suggest that some of these residues are rescued more than expected and perhaps alter signaling and/or internalization. Are these residues thought to interface with the G protein? The internalization aspect is also interesting. We'd be curious to know how the authors might interpret any interplay of internalization with the surface expression measurements, especially in the context of variants that might increase basal signaling activity of the receptor.

Line 392-398. The authors describe mutation effects in sites that would bias the ensemble in different ways that could effect drug binding. This is premature and doesn't have a good experimental basis. For instance, backing up these claims would likely need to have experimental measurements of dynamics such as smFRET or MD simulations which have not been done on these mutants. Perhaps just describe the effects in the context of the available knowledge and not reach towards a model of receptor dynamics.

Line 420. The authors claim temperature rescue proves that variants are disrupting 'thermodynamic instability'. As described throughout this review - temperature rescue is complex and there is a deep literature on the effects of mutations. We recommend the authors contextualize the results with existing literature. The results here are completely in line with expected results from the field and the way this discussion section is written it sounds like none of these studies have previously been done.

Line 429. The authors state that in contrast with prior studies on CFTR and Rhodopsin that in this work they find 'universal rescue' as if it disproves previous models of chaperone function. These are different proteins and in the context of the CFTR studies there were multiple chaperones to compare chaperones effects directly. While a beautiful study, this study does not disprove existing models on a completely different protein. This should be acknowledged more explicitly in the discussion.

Line 440. The authors describe the energetic effects of binding a drug as overcoming the effects of mutants. However, earlier the authors describe the effects of mutants in biasing the ensemble. We agree more with this framing rather than the

ensemble based explanation which is partially why we caution the interpretation that mutants alter the conformational ensemble.

Line 442. The authors describe that chaperones may rescue variants in a more general manner than was previously considered. Pharmacochaperones are widely regarded as a method to rescue expression in proteins, and many companies are currently chasing making them for many membrane protein associated diseases. We don't think this framing is overstating the novelty of the finding and does not accurately represent the state of the field.

Line 448 the authors try to contextualize the results on drugs as the mostly additive effects of mutants. This section will be non-obvious to the vast majority of membrane protein people who will be interested in this paper. We encourage the authors to unpack these ideas a bit. What ideas are similar? Why will the effects be similar? How does the law of mass action apply here? Overall, this paragraph is confusing and complex and we recommend unpacking the ideas and explaining them more.

Line 464. The authors refer to similar experiments as a high throughput protein abundance selection experiments and in the next sentence describes that as shown in this sentence these assays can be useful in exploring chaperones. This is a bit confusing as the prior studies were yeast-based growth based selections whereas this is a surface expression screen. Please adjust the language accordingly.

Last paragraph. The authors try to extend that because they find a single ligand rescues nearly all loss of surface expression mutants than any ligand that binds to the receptor could rescue variants. This is an over-reach. If they wanted to make that claim they would need to show many new ligands that bind (including structures or SPR or something like that) all over the receptor that they rescue the same mutants. As the authors have shown the results for a single previously characterized ligand... While the ideas here are interesting, they are not backed up by the manuscript.

Data availability: We strongly encourage the authors to upload the DMS data to MaveDB.

By Matthew Howard and Willow Coyote-Maestas

Reviewer #3 (Remarks to the Author):

Protein misfolding has been implicated in the molecular basis of numerous genetic diseases, and the recent development of small molecule pharmacochaperones that suppress misfolding reactions has revolutionized the treatment of several such diseases. In theory, mechanistic basis for the activity of these molecules is quite simple- ligands that selectively bind to the native fold should enhance folding via mass action. However, in the ~ 20 years since the conceptual development of such molecules by Jeff Kelly and colleagues, it has become quite clear that the stabilization generated by such molecules can play out in nuanced ways in the context of the cell. For instance, while pharmacochaperones that target the CFTR protein have revolutionized the treatment of cystic fibrosis, the best molecules and combinatorial therapies that are currently available in the clinic are only effective towards certain clinical CFTR variants. Ongoing work to understand how these molecules do or don't rescue misfolded variants and how we can maximize their efficacy towards diverse clinical genotypes is an important challenge in precision medicine. In this work, Mighell & Lehner investigate how an existing drug that acts as a pharmacochaperone (Tolvaptan) modulates the expression of a comprehensive library of ~7k variants of the vasopressin-2 receptor (V2R), the misfolding of which causes nephrogenic diabetes insipidus (NDI). While several previous reports have demonstrated that select misfolded clinical V2R variants can be rescued by pharmacochaperones, this work provides a more comprehensive profile of the expression pattern, temperature sensitivity, and Tolvaptan response of thousands of random missense variants. Their findings show that about half of the clinical NDI variants reduce V2R expression, and that most of those can be rescued, to some extent, by Tolvaptan. This is a wonderful, impressive, and technically sound data set, and the manuscript reports a variety of interesting findings. However, while the writing is quite clear, I find the overarching interpretations to be overstated and several of their conclusions to be somewhat misplaced. There are also several aspects of the manuscript that could benefit from clarification. I have summarized my suggestions as follows:

Conceptual Issues:

-Overall, the manuscript seems to imply that the idea that pharmacochaperone activity should be generalizable to a variety of misfolded variants is a novel feature of this manuscript. For instance, the authors state "Here we use V2R as a model system to directly test the hypothesis that PCs can rescue destabilizing mutants across the structure of a complex human protein." I appreciate that there are a fair number of cellular biologists that view the effects of these compounds from a phenomenological/ empirical viewpoint, and think these points are worth re-stating in the plainest of language until this perspective is widely appreciated. Nevertheless, I think many biochemists and biophysicists within the proteostasis community have made these points many times over the years (see doi: 10.1146/annurev.biochem.052308.114844 and 10.1016/j.sbi.2021.11.009). In fact, in light of the physical arguments made by these authors and many others, the mere fact that all variants do not respond to such compounds is perhaps the most interesting part. Based on these considerations, I think the fact that the authors have described this paper as a proof of concept for the fact that some pharmacochaperones can rescue lots of mutants seems a bit odd and inappropriate. It doesn't stress the most interesting parts of the manuscript either, in my opinion.

-As the authors have noted, several other pharmacochaperones have been characterized against variants of several other target proteins and few, if any, have demonstrated such a broad efficacy towards such a diverse range of mutants as

Tolvaptan. The authors imply in the discussion that the previous studies on this topic have somehow missed the point because they didn't look at enough mutants, noting "the failed rescue of individual variants (led) to suggestions of widespread idiosyncratic effects between PCs and variants." But this seems like strange logic. The fact that these authors have identified such widespread rescue for a different protein with a different pharmacochaperone doesn't mean the previous studies are somehow invalid or misguided. It is certainly possible (and perhaps likely) that if previous investigations of other targets had profiled thousands more variants, they may have identified significantly more mutants that can be corrected by these other pharmacochaperones. However, collecting more measurements for random mutations and using them to train new models won't change the fact that the "idiosyncratic" clinical mutations focused on by these investigations don't respond to the drugs in question- the most important pharmacological issue. Given the divergent observations the authors have made for random mutations relative to more focused investigations of disease mutants, it seems more appropriate to consider whether this is a unique property of this target receptor relative to other targets. Or perhaps the disease mutants in V2R differ from those of CFTR or other proteins? Or perhaps there is some feature of this specific compounds that endows it with broader efficacy relative to others (i.e. large binding energy). From my experience with CFTR modulators, I can tell you this later point seems most plausible, as different ligands with similar binding energy can have strikingly different effects- an observation that cannot be explained by thermodynamics. Indeed, cells are not at equilibrium, and most biochemical processes are subject to kinetic control, including the coupling between folding and trafficking in the ER. Energetics are highly useful, but often do not tell the whole story. I feel like this is a more useful discussion to have for the field.

-The authors state: "To maximize the effectiveness of PC therapy though, it is necessary to identify the mechanism of all pathogenic variation for a given protein, as well as the response to PC. These data do not currently exist for any protein" I think this is an overly strong statement. I think recent work on CFTR is perhaps the most comprehensive to date (doi: 10.1016/j.jcf.2024.02.006). While this study does not report 7k random variants, it includes all 655 currently known pathogenic mutations and multiple pharmacochaperone treatments. Importantly, this investigation also uses techniques that are accepted by the US FDA for the expansion of the therapeutic label of these drugs, which is something that DMS unfortunately cannot be used for currently. It is also worth noting that a comprehensive (albeit small) library of pathogenic rhodopsin mutants has been used to profile five different pharmacochaperones to date (doi: 10.1016/j.jbc.2022.102266, 10.1093/hmg/ddac125, and 10.1371/journal.pbio.3002932). Though these libraries are an order of magnitude smaller than the library of random variants characterized in this work, they are arguably more complete from the perspective of clinical genetics, as they include indels that are found in the CF and RP patient population. The omission of these variants from missense libraries is not a trivial issue, as indels are quite common in the clinic and can be highly penetrant (ie $\Delta F508$ CFTR).

-The authors state "the stabilization conferred by small molecule binding should be largely independent of where a small molecule binds and where a mutation is located, provided the compound specifically binds the native folded state." While we agree this should be the case, the authors have noted several cases in the literature that show that empirically, this is not actually the case (see refs 14, 36, and 68). Again, these observations are not invalid simply because a larger number of other random mutants were not included in these libraries. The regiospecific variations and the distinct effects of pharmacochaperones that bind to different sites would still be there, regardless of the library size.

-The authors note "Namely, the magnitude of rescue is greatest for variants with starting free energy values in the steepest part of the free energy- expression level curve (Fig. 3d)." I agree with the authors interpretation of this observation. However, the authors should note that similar observations have been previously made for both CFTR and rhodopsin variants (see Refs 67 and 68). The authors cited these papers to note they used a relatively small library, but failed to mention that both of these studies made similar observations and offered this same explanation previously. It is instead offered as a novel finding made in this study. It is worth noting that the sensitivity of the moderately expressed variants seems to be an emergent phenomenon that appears to apply to most targets. I should also note that, had the authors applied the thermodynamic bounding analysis outlined in refs 67 and 68, I suspect they would find the trends in Tolvaptan response are relatively similar to those observed for CFTR and rhodopsin- many respond but not to the extent that the thermodynamic effects of ligand binding would suggest they should. This phenomenon is not so much a yes/ no question, as the authors manuscript suggests, but rather more a question of how much. Discussing which aspects of this phenomenon are the same or instead differ across target proteins is much more useful than litigating methodological shortcomings.

Technical Issues:

In their description of how they calculated variant scores, the authors note that read counts were "multiplied by the geometric mean fluorescence value associated with each bin." Could the authors please explain why they chose to use the geometric mean instead of the arithmetic mean? I am aware that many in the flow cytometry community still use geometric means. However, it is my understanding that this is a relic from a bygone era- geometric mean was previous used when cytometers had few fluorescence channels and intensity measurements were collected using log scale bins to avoid compressing dynamic range. The geometric mean of log data transforms it into a linear mean, which overcame the limitation of collecting logarithmic data. But these days, most cytometers have 256k channels and collect data using linear bins. Is there a reason for using a geometric mean in this case? Will it improve variant scoring? I am genuinely curious.

-The authors provide little detail on how variant scores are binned into the poorly expressed, moderately expressed, and well-expressed classifications. Were the boundaries indexed to some reference (ie WT or a misfolded mutant)? I recognize that this is an arbitrary detail that will likely not affect the outcome, but I know that I will not be the only reader to consider this issue. It might be good to add a brief statement on the criterion and/ or provide a supplemental analysis showing the conclusion is relatively insensitive to the positions of the bins.

-By DMS standards, the authors note admirable precision between replicates in the supplement. However, the effects of Tolvaptam on many variants is relatively subtle. In this regard, the authors use of a larger library has its own tradeoffs, such as lower signal to noise due to dilute sampling. In evaluating the reported scores, it seems that for many variants (if not most), the difference between the scores in the presence and absence of drug is small relative to the SEM of each measurement. There is little discussion of the precision of the shifts and, indeed, the variants of interest in Fig. 4i depict error bars without even noting what they correspond to. The authors should provide an analysis, complete with error propagation, that states how many of the observed shifts are statistically robust. I don't mean to quibble, but such considerations will be important for the design of future DMS screening approaches, where the comparison of subtle drug effects on specific variants could be highly consequential for decision making that may occur during the drug development process.

Overall, I would like to re-affirm that this is an excellent study that deserves to be highlighted in this prestigious journal. I just think the authors should first make an earnest effort to bridge this study with other emerging observations rather than focusing on the potential shortcomings of previous methodologies. There is more to be learned from the nuance within their data than can be learned from glossing over the outliers. I genuinely hope the authors find these comments useful. With improvements, this is sure to be a landmark paper.

Version 1:

Decision Letter:

Our ref: NSMB-A50265A

29th Apr 2025

Dear Dr. Lehner,

Thank you for submitting your revised manuscript "A pharmacological chaperone stabilizer rescues the expression of the vast majority of pathogenic variants in a G protein-coupled receptor" (NSMB-A50265A). It has now been seen by the original referees and their comments are below. The reviewers find that the paper has improved in revision, and therefore we'll be happy in principle to publish it in Nature Structural & Molecular Biology, pending minor revisions to satisfy the referees' final requests and to comply with our editorial and formatting guidelines.

We are now performing detailed checks on your paper and will send you a checklist detailing our editorial and formatting requirements in about 2-3 weeks. Please do not upload the final materials and make any revisions until you receive this additional information from us.

Sincerely,

Katarzyna Ciazynska, PhD
(she/her)
Senior Editor
Nature Structural & Molecular Biology
<https://orcid.org/0000-0002-9899-2428>

Reviewer #1 (Remarks to the Author):

I have read the revised manuscript in full, and only have a few remaining minor comments:

- 1) Lines 173 to 177: The manuscript mentions "The predicted free energy change (ΔG) of membrane integration for these helices is relatively unfavorable", and "TM7 is predicted to be inefficiently incorporated in the membrane", but it's hard to put too much weight into these statements without seeing some corresponding data / values.
- 2) Lines 314 (and elsewhere, later on): Should Lowess be capitalized to LOWESS, since it's an acronym?
- 3) Line 383: The GPCRdb superscripted numbering is first used here, but first defined on line 408. I suggest moving the parenthetical definition up to this line.

Reviewer #2 (Remarks to the Author):

The authors did a fantastic job addressing the all the concerns I had either by making adjustments in the text, the figures, or explaining their rationale. Kudos for a beautiful paper!

Willow Coyote-Maestas and Matthew K. Howard

Reviewer #3 (Remarks to the Author):

The authors have done an adequate job of addressing the many lengthy requests made by the reviewers. I am grateful for their careful consideration and have no further requests.

Version 2:

Decision Letter:

24th Jul 2025

Dear Dr. Lehner,

We are now happy to accept your revised paper "A small molecule stabilizer rescues the surface expression of nearly all missense variants in a GPCR" for publication as an Article in Nature Structural & Molecular Biology.

Your paper will be published online soon after we receive proof corrections and will appear in print in the next available issue. You can find out your date of online publication by contacting the production team shortly after sending your proof corrections.

If you have not already done so, we strongly recommend that you upload the step-by-step protocols used in this manuscript to the Protocol Exchange. Protocol Exchange is an open online resource that allows researchers to share their detailed experimental know-how. All uploaded protocols are made freely available, assigned DOIs for ease of citation and fully searchable through nature.com. Protocols can be linked to any publications in which they are used and will be linked to from your article. You can also establish a dedicated page to collect all your lab Protocols. By uploading your Protocols to

Protocol Exchange, you are enabling researchers to more readily reproduce or adapt the methodology you use, as well as increasing the visibility of your protocols and papers. Upload your Protocols at www.nature.com/protocolexchange/. Further information can be found at www.nature.com/protocolexchange/about.

Authors may need to take specific actions to achieve compliance with funder and institutional open access mandates. If your research is supported by a funder that requires immediate open access (e.g. according to [Plan S principles](https://www.springernature.com/gp/open-science/plan-s-compliance) or the [NIH public access policy](https://www.springernature.com/gp/open-science/us-federal-agency-compliance)) then you should select the gold OA route, and we will direct you to the compliant route where possible. Because authors warrant under our subscription licensing terms that they haven't committed to licensing any version of their article under a licence inconsistent with the terms of our agreement – including the applicable embargo period – publication under the subscription model isn't suitable for authors whose funders require no embargo.

Sincerely,

Katarzyna Ciazynska, PhD
(she/her)
Senior Editor
Nature Structural & Molecular Biology
<https://orcid.org/0000-0002-9899-2428>

Reviewer #1 (Remarks to the Author):

In this manuscript, the authors perform and analyze the results of a cell-surface trafficking assay for missense variants of the vasopressin 2 receptor (V2R) protein in cultured human cells. The authors first use a well-established Sort-Seq approach to sort cells each expressing individual members from a large library of V2R single amino acid variants based on fluorescent antibody binding, indicative of the amount of variant V2R protein that made it to the cell surface. The authors repeat the experiment at reduced temperatures and pre-incubate with the V2R antagonist, tolvaptan. The authors identify that the vast majority of V2R loss-of-function variants are caused by presumed protein destabilization, and that cell-surface expression of these proteins can be rescued by tolvaptan, with clear implications for pharmacological chaperones (PCs) as a generalizable means to rescue destabilizing variants in membrane proteins. The main feature of this paper is that the authors provide a significant advancement in potential clinical considerations for PCs. While the experiments and the ensuing analyses appear well performed, the manuscript suffers from exaggerative language that overstates the potential impacts of the results, which need to be revised prior to publication.

We thank the referee for their enthusiasm and very constructive suggestions.

Major comments:

1) It's unclear what the generality of the PC phenomenon is across protein targets and putative PCs. Only in the discussion do the authors introduce similar, albeit smaller-scale studies performed with Rhodopsin (another GPCR!) and CFTR, that seem to support the alternative hypothesis of PCs being far more selective in the proportion of destabilizing missense variants it rescues. These studies should be mentioned much earlier in the manuscript, so that the potential universality of the results is better contextualized and understood by the reader. These differences are particularly stark, as all studies (including a recently published manuscript in PLoS Biology on Rhodopsin cell-surface trafficking rescue by PCs; PMID: 39808594) seem to utilize very similar transgenic overexpression systems, albeit with different gene targets and candidate PCs. Do the authors have any ideas as to why they observed such large-scale rescue of variant trafficking for V2R with tolvaptan, while the other authors observed more region-specific rescue of variants with their protein and drug targets? Could it be the location of small molecule binding? Or could it be related to the amount of stabilization conferred by small molecule binding, which will drastically differ depending on the protein variant target and small molecule in question? This information would seem to help the reader understand whether the effect observed with Tolvaptan and V2R cell surface rescue is expected to be an unusual outlier case, or whether it's likely to be more commonplace across contexts.

Thank you for these comments – we agree. We have modified the introduction to more clearly present the possibility that some PCs may have specific-rescue effects, while some (as Tolvaptan for V2R) may have largely generic rescue effects (changed text in bold):

*Depending on the mechanism of action, **some** PCs could have high specificity, rescuing only subsets of pathogenic variants localized in particular regions of the protein^{13,14}. Alternatively, **some** PCs could **behave simply, in accordance with the law of mass action, and have largely nonspecific stabilizing effects that offset the destabilization by most variants in a protein, wherever they are located**^{12,15}.*

We have also expanded the discussion:

*Previous low-throughput studies mostly tested the efficacy of PCs for a small number of variants, with the failed rescue of individual variants leading to suggestions of widespread idiosyncratic effects between PCs and variants^{32–35}. More systematic efforts testing hundreds of variants in rhodopsin^{14,54} and CFTR⁵⁵ demonstrated rescue of many variants, but with the authors also suggesting substantial region-specific differences in rescue. **The rhodopsin investigations emphasized differences in rescue between mutations in transmembrane helices 2 and 7¹⁴, but subsequently found that 67 out of 69 reduced expression retinopathy variants had measurably increased expression in the presence of a PC⁵⁴, consistent with the findings here. CFTR has multiple folding domains⁷², and PCs were found to be most effective for variants in close proximity to the PC binding site⁵⁵. Class A GPCRs then, as single-domain proteins, may represent amenable targets for PCs. We have profiled here an order of magnitude more variants than these previous studies to demonstrate protein-wide, nearly universal PC rescue of variants.***

2) What is the relationship between surface expression and function? This method measures surface expression, but the authors don't directly show any evidence showing that rescued trafficking correlates perfectly with rescued function. This is particularly important for the Tolvaptan PC experiments. Tolvaptan is described in the manuscript as a “selective antagonist”, with a hypothesized binding site overlapping the binding site for the hormone arginine vasopressin (AVP). A quick literature search seems to suggest that it indeed counteracts the signaling pathway normally initiated by AVP binding, thus pharmacologically conferring a loss of function to V2R. Thus, it would seem that while tolvaptan may bind to V2R and help rescue its cell surface localization, that its constant presence may also interfere with the receptor's ability to bind its endogenous ligand and transmit signal, thus not functionally rescuing the

variants. It would seem that NDI-associated variants, which are currently largely loss-of-function through destabilization and mistrafficking, may simply become properly trafficked but still non-functional variants in the presence of tolvaptan. Experimental evidence demonstrating functional rescue, or citations demonstrating the functional rescue of endogenous AVP signaling upon co-incubation with Tolvaptan, seem necessary to support any potential functional implications of rescue. Relatedly, the potential for tolvaptan to inhibit WT copies while rescuing the cell surface trafficking of destabilized variants seem to potentially impact prospects for using this drug in NDI affected females.

These are all very good points. It is true that Tolvaptan is an antagonist, and that its presence would be presumed to interfere with the receptor's ability to bind its endogenous ligand and transmit signal. While extending this study to also measure V2R signaling would be very valuable, it would require an entirely different experimental setup. In lieu of this data, we do provide citations that support the notion that a substantial fraction of V2R variants would be able to signal at the membrane:

Tolvaptan (also known as OPC-41061) is a V2R-specific, competitive, small molecule antagonist⁵⁷ (Fig. 4b), and is approved for treating autosomal dominant polycystic kidney disease⁵⁸, in which AVP-V2R signaling is misregulated. However, Tolvaptan has also been explored in vitro for its activity as a PC. In most tested cases, Tolvaptan not only rescues surface expression, it also enables some level of AVP-mediated signaling^{33,35,59,60}.

The interpretation is that, while Tolvaptan will remain bound throughout the biosynthetic process, it has some off-rate that means it will eventually dissociate from the receptor, leaving it free to bind endogenous ligand.

In particular, citation 35 (<https://pubmed.ncbi.nlm.nih.gov/16926443/>) uses competition binding assays to show that V2R mutants in cells pre-treated with Tolvaptan actually increase the number of AVP (endogenous ligand) binding sites by two to three fold.

Additional evidence comes from citation 56, in which another small molecule (SR49059) was actually tested in a clinical trial for NDI. While this is a different molecule from Tolvaptan, it has the same pharmacology (i.e. orthosteric antagonist). This clinical trial actually saw improvement in core symptoms of NDI (indicating effectiveness of the PC therapy) but was discontinued due to off-target side effects.

Regarding the possibility of Tolvaptan inhibiting WT copies- this is one reason why V2R- NDI is an attractive disease for PC therapy- the affected individuals are almost exclusively hemizygous males- in other words, there is only the mutated copy of V2R.

3) Disconnect and inconsistency in strength of conclusions made in the

manuscript: eg. in the abstract “Strikingly, treatment with a PC rescues the expression of nearly all destabilized variants, with non-rescued variants identifying the drug’s binding site” while in the results section “The few sites where variants are consistently not rescued inform on the likely binding site of Tolvaptan as well as other functional sites of the receptor”. The latter statement seems more appropriate, unless experimental evidence showing drug binding at that location is provided. With nearly the entirety of the work and analysis resulting from data from a singular assay (repeated in different conditions), I do think it is generally more prudent to leave many of these interpretations outside of cell-surface localization as possibilities or likely implications, rather than definitive statements.

We agree that the statement in the abstract was too strong; we have clarified:

*The non-rescued variants identify the drug’s **predicted** binding site.*

Minor comments:

4) Yellow outline visualization scheme in Fig 5 is really hard to see.

Thanks for pointing this out: we have changed the highlight color to gold and changed the lighting representation of these structural figures, which makes the outlines easier to see.

5) Line 326-328: confusing grammar; would suggest rephrasing for clarity.

We changed this sentence to read:

Based on the broad effectiveness of temperature rescue, we predicted that PC binding could also have a general rescue effect.

Reviewer #2 (Remarks to the Author):

Summary:

Mighell & Lehner conduct a series of deep mutational scanning experiments on the Vasopressin 2 Receptor (V2R) where they investigate mutational effects on receptor surface expression in the presence of reduced temperature or a pharmacochaperone (PC). They show that both reduced growth temperature (27C) and PC (tolvaptan) treatment can improve expression of normally poorly-expressing variants. Results are interpreted in the context of known pathogenic human variants, structural data, and a variety of thermodynamic and other variant effect predictors. Overall this work increases our understanding of sequence-to-expression determinants in GPCRs and the mechanism of temperature and PC chaperone effects. That said the manuscript would be greatly improved through a major revision by putting the work in context of past work in the membrane protein biogenesis field and reducing the generalizations made based solely on this experiment on a single receptor and single pharmacochaperone vs temperature.

We thank the referees for their enthusiasm and very constructive suggestions.

Major points

Strengths

The authors embark on the first of its kind experiment measuring the how temperature and a small molecule chaperone rescue all mutations in a membrane protein, V2R. As small molecule chaperones have transformed the treatment of cystic fibrosis and many in industry are embarking on similar campaigns in other membrane proteins, understanding the mechanisms of how chaperones rescue mutations will be broadly useful. Similarly, that temperature rescue was done here can begin to answer questions on the mysterious nature of temperature rescue.

Membrane protein biogenesis is an important and conventionally intractable area of study. The authors add a critical new mutational scanning dataset that can help understand how mutations break membrane proteins and in turn how residues contribute to surface expression.

Opportunities for improvement

In the first line of the abstract authors say reduced protein stability is the most common effect of mutations. I think while it is true that protein abundance is the most common effect of mutations, stability is a specific

property referring to the life-time of the protein at least in the context of membrane proteins. We believe this confusion is likely coming because in the protein folding space - fold stability is used as a catch-all term.. However, for membrane proteins where mutations can alter biogenesis (which is a catch-all term to mean the formation of protein) vs a protein's stability - this is an inaccurate statement. Probably better just to use the word 'abundance'. For instance, if the authors are curious to know where these classifications come from we recommend they read the CF variant classification literature where biogenesis and stability are distinct mechanisms and most of the pharmacochaperones here are acting upon biogenesis likely but without a distinct measure that separates these it's hard to differentiate the effects. In general, I would recommend the authors change the language throughout the manuscript to represent this nuance because while pedantic - this paper will be read by the membrane protein biogenesis/stability/folding field and these terms must be referred to specifically. For instance the citations the authors use were not measured in the context of differentiating the formation vs the degradation of proteins - so that statement is not backed up by the literature.

Thanks for this suggestion – we agree. We have changed “stability” to “abundance” throughout the text (changes in bold):

Abstract:

*Reduced protein **abundance** is the most frequent mechanism by which rare missense variants cause disease. A promising therapeutic avenue for treating reduced **abundance** variants is pharmacological chaperones...*

Introduction:

*The most frequent mechanism by which missense variants cause rare diseases is reduced protein **abundance**. Large-scale experimental⁵ and computational^{6,7} surveys estimate that 40-60% of pathogenic variants are explained by loss of **abundance**. Compensating for this reduced **abundance** therefore represents a potentially general strategy to treat rare diseases.*

Discussion:

*Previous experimental⁵ and computational^{6,7} approaches estimate that 40-60% of pathogenic variants are explained by loss of stability or **abundance** (which is in line with our findings here)...*

Currently, the figures seem to have many many panels and be full page figures. For instance the first figure has many comparisons that may not need to be in the main figure such as the membrane insertion indices or the hydrophobicity scores. Similarly, figure 2 has many different

comparisons of pathogenic predictors - these sorts of studies have been done previously and I did not learn much from these panels. Figure 3, similarly seems to take up a tremendous amount of space with concepts of how a LOWESS fit works. In general we recommend the authors work on focusing the figures and panels on the observations and conclusions they most want to highlight in the manuscript.

To clarify, Figure 3e isn't explaining how Lowess fit works, it's schematizing our approach for using a Lowess model to subsequently find residues with significantly deviating rescue behavior. We believe this is useful and, in our experience, important for communicating the approach to a general audience.

The most interesting and novel part of this study is measuring how reducing the temperature and adding pharmacochaperones for all possible mutations effect expression of V2R. However, the authors don't compare between these these two conditions. These are really different mechanisms of rescue and we think it would be worth spending more space directly comparing the two and exploring how they may be different. Are the positions rescued more or less different between the two conditions? In general, temperature based rescue of expression is a bit mysterious as it's unlikely to be due to thermodynamic stability and could more be due to slowing down the processes of biogenesis and therefore increasing the fidelity of surface expression. In contrast the binding of a pharmacochaperone is likely to stabilize specific states in biogenesis. If the residues that rescue both are the same but just a difference in magnitude that is worth explicitly discussing. If not, it also very interesting and worth highlighting.

Regarding thermodynamic stability- we don't agree that temperature-based rescue of expression is 'unlikely to be due to thermodynamic stability'. The stabilization of proteins by temperature reduction is extremely well established experimentally and theoretically. It seems much more speculative - and less parsimonious - to invoke a model of specific interactions with biosynthetic machinery.

We note that citation 14 is a paper by Jon Schleich invoking the exact same simplifying framework we use here:

[from citation 14] "Temperature sensitivity is typically associated with unstable variants that are poorly expressed, and it is generally assumed that their enhanced expression arises from an increase in thermodynamic stability and a corresponding increase in the fraction of folded protein at reduced growth temperatures."

However, for clarity, we have changed the header (change in underline):

Temperature rescue of V2R variants

Also, we add a sentence to explicitly identify that in some cases there may be other mechanisms of rescue, and cite the 1992 CFTR paper:

*In principle, variants that are poorly expressed due to decreased thermodynamic stability should be rescued by incubating the cells expressing the variant library at reduced temperature (Fig. 3a). **While reduced temperature could also affect biosynthetic processes that result in increased expression⁵³, we assume that in the majority of the cases rescue is via thermodynamic stabilization¹⁴.***

And, we make these changes:

*We sought to understand what fraction of V2R variants could be rescued by **temperature reduction** by culturing the cells expressing the V2R variant library at 27°.*

*This suggests that the majority of variants can be rescued by **temperature reduction**.*

***Based on the broad effectiveness of temperature rescue**, we predicted that PC binding could also have a general rescue effect.*

Regarding comparison between 27° and Tolvaptan rescue- good point- we have now included an analysis of 27° rescue with Tolvaptan rescue (Supplemental Figure 3i).

We also include this new text in the results:

Of the 10 outlier residues for 27° rescue, 6 are also outliers for Tolvaptan: S5, T6, S167^{4x53}, are rescued less than expected in both cases, and L336^{8x54} is rescued more than expected in both cases. Interestingly, while S24 and Y205^{5x39} are rescued more than expected in 27°, they are actually less rescued than expected in Tolvaptan condition. Overall, though, rescue in the two conditions is correlated ($\rho = 0.49$, Supplemental Fig. 3i).

In the discussion, the authors present the work as if it is contradictory to the field and prior results. While this study is the first study of how a single pharmacochaperone rescues all single mutations in V2R, there have been many previous studies on the mechanisms of other pharmacochaperones in other membrane proteins. The authors discuss the work as if pharmacochaperones (based on their work) are a novel approach for curing disease, but this has previously been established. The authors seem to argue that most mutants are rescued by a single chaperone implies that prior results on CFTR in which different pharmacochaperones rescue different subsets of mutations is wrong. While this is an interesting example, the authors have not sufficiently tested this hypothesis and do not have other compounds to compare to be able to be convincing. It is perhaps unsurprising that different protein

architectures (GPCR vs transporter) have different results. We encourage the authors to consider not over-weighting the results on this single protein with a single compound to override results in completely different proteins that include structural characterization and more biophysical measurements.

We agree and appreciate this point, raised by multiple reviewers. See the updated text in discussion here:

Previous low-throughput studies mostly tested the efficacy of PCs for a small number of variants, with the failed rescue of individual variants leading to suggestions of widespread idiosyncratic effects between PCs and variants³²⁻³⁵. More systematic efforts testing hundreds of variants in rhodopsin^{14,68} and CFTR⁶⁹ demonstrated rescue of many variants, but with the authors also suggesting substantial region-specific differences in rescue. The rhodopsin investigations emphasized differences in rescue between mutations in transmembrane helices 2 and 7¹⁴, but subsequently found that 67 out of 69 reduced expression retinopathy variants had measurably increased expression in the presence of a PC⁶⁸, consistent with the findings here. CFTR has multiple folding domains⁷⁰, and PCs were found to be most effective for variants in close proximity to the PC binding site⁶⁹. Class A GPCRs then, as single-domain proteins, may represent amenable targets for PCs.

Figure 6 gives a misleadingly simple thermodynamic model of the 'folding' of V2R. Membrane protein biogenesis is complex and the experiments done here do not give sufficient information to propose such a model. For instance, the authors are measuring surface expression - while it is likely most of the effects of a mutation are due to biogenesis or stability - the authors do not necessarily know which variants may just be affecting surface expression. We recommend removing this completely and ending on figure 5.

Of course, as all models are, we simplify the situation in Figure 6. However, the model we present is parsimonious and very consistent with the data. We believe it is the correct working model that should be used in the absence of additional data refuting it for particular variants. We agree that it is likely that other proteins (for example, CFTR, with distinct folding units) may behave differently than V2R.

Minor Points:

Line 102: The authors state they test the hypothesis that pharmacochaperones can stabilize mutations across a 'complex protein'. While this study is novel in that it measures the effects of all mutations, Jon Schleich's group has previously tested this hypothesis in several hundred mutations that are distributed across CFTR. More recently the cystic fibrosis foundation tested ~700 mutations in laborious assays that confirm this result ([https://www.cysticfibrosisjournal.com/article/S1569-1993\(24\)00021-3/fulltext](https://www.cysticfibrosisjournal.com/article/S1569-1993(24)00021-3/fulltext)). We recommend the authors change language

that don't claim to have discovered this for the first time. Further, it's unclear what a 'complex protein' is here - perhaps be a bit more specific.

We have re-written this sentence which now reads:

Here we use V2R as a model system to directly test whether PCs can rescue all destabilizing mutants in a protein.

Line 110: Similarly, here the authors say their results provide proof that a pharmacochaperone can rescue mutations across the receptor - this is well-known from the same CF literature as the prior comment.

We have deleted this sentence.

Line 149: We found the normalization of using synonymous and nonsense variants confusing here but this is probably my reading. Is it that all nonsense and synonymous variants are set at 1 or 0 or is it that the distribution is set? Reading the corresponding section of the methods didn't really clear up my confusion. More detail would be useful. Has this approach been used previously? It seems there is no literature cited here but many others have done FACS based screens - it would be useful to understand whether this is what others use for similar experiments.

As there is one wild-type sequence (here not including synonymous variants), we use this one genotype to define surface expression of 1. As there are many different presumed total loss of function variants (i.e. nonsense variants), we used the median of these to define 0.

Words added to methods to clarify:

*Then, these surface expression scores are normalized such that **the** wild-type **genotype** = 1 and the median of known loss-of-function variants (premature stop codons before the 300th residue) = 0.*

This normalization scheme has indeed been used before, for example in the VAMP-seq paper: <https://pubmed.ncbi.nlm.nih.gov/29785012/>

Line 169: The authors state that previous work described TM3 plays a critical role in receptor stability across all class A GPCRs. However, the papers they cite are computational models and have little experimental basis. There have been a number of DMS experiments done on GPCRs and extensive studies of biogenesis and stability in rhodopsin - in general it doesn't seem that this model has been proven true across these datasets.

We don't want to litigate the accuracy of methods or assertions in other studies; while the transmembrane helices may have somewhat different sensitivities across the GPCR family, our findings are in accordance with the conclusions made by the cited sources.

Line 174: The authors compare mutational effects to membrane insertion indices qualitatively. In Figure 1F TM7 has a higher insertion ΔG than TM3 but the authors state that TM3 has amongst the lowest expression and that maybe is because of the insertion index. I find this incongruency a bit confusing. Perhaps either directly compare the two quantitatively or don't include the membrane insertion indices and parse TM scores in 1E altogether.

We agree that it's incongruent not to mention TM7 since it has the highest predicted ΔG of insertion. We have added text to include this:

*The predicted free energy change (ΔG) of membrane integration⁴² for these helices is relatively unfavorable suggesting that mutations here could further compromise an already inefficient process. **In contrast, TM7 is predicted to be inefficiently incorporated in the membrane, while mutations here are well tolerated (Fig. 1f).***

Lines 176-185: The authors compare the effect of mutations that reduce hydrophobicity. I think some explanation of why this analysis was done would be worthwhile here. Furthermore it is expected that residues facing the membrane vs those that don't will have different effects. Perhaps the authors should compare this.

Indeed, we introduce the premise in the main text before presenting the analyses:

TM regions are solvated in lipid and are therefore enriched in hydrophobic residues, compared with extra- or intracellular residues

Supplemental Figure 1f shows that membrane pointing positions prefer to be hydrophobic, while many interior-facing residues tolerate hydrophilic amino acids.

Line 186 The authors compare the effects of mutations in glycosylation sites. The authors describe 'strong' mutational effects - what is a strong effect? Is this similar to mutations within membrane facing residues? In some ways the alternative explanation here is that any single glycosylation is not critical yet they combined together effect expression. Furthermore, we suggest caution in making these claims especially in the context of an over-expression system in which the assay may not be able to detect the effect of removing a single glycosylation site.

We agree that "strong" is vague and have added in the median score at each position. We also agree that more qualifying language should be used to specify that these findings can only be applied to this specific context:

Positions 22-24 represent the N-glycosylation motif (N-X-S/T), and mutations at position 22 and 24 are much more deleterious than their neighbors in the unstructured N-terminus (median position score=0.77 and 0.68 respectively, Supplementary Fig. 1d). While O-glycosylation was reported at several serines and threonines in the N-terminus⁴², we only see strong mutational effects at S5 and T6 (median position score=0.74 and 0.73, respectively), suggesting these are the only critical glycosylation sites, at least in HEK cells with this overexpression system.

Line 200: The authors compare surface expression to pathogenic classifications. There are no statistical tests here - we suggest the authors add these. Related, figure 2A and 2B seem slightly redundant, perhaps consider condensing.

In lieu of a statistical test comparing the distributions of pathogenic and benign variants, we decided to perform an AUROC analysis (Figure 2e), which we think gives a better intuitive sense of how well surface expression discriminates between pathogenic and benign variation than a p-value. We think figure 2a and 2b are not redundant- 2b helps the reader to immediately understand which fraction of variants in each classification are in which expression category. And, since we use those categories throughout the paper, we think it makes sense to use them here as well.

Figure 2d - The authors label VEP plus mechanistic screen. We feel calling this a mechanistic screen is a bit much as there is no readout for activity. This is evidenced by the statement in line 253 of 'presumed signaling defect'. We caution against the use of this term here.

The rationale for calling this a mechanistic screen is in contrast to VEPs, which are a convolution of all potential mechanistic effects. The point here is that we can dig down one level below VEPs to at least classify the mechanism as either loss of abundance or not; and 'not' in this case would be presumed to have a defect in signaling.

Figure 2H-J captions are missing from the legend.

Thanks for catching this omission. We have added captions:

h NDI variants of different expression levels illustrated on the V2R structure (PDB ID: 7KH0). *i* Surface expression of all variants predicted by AlphaMissense to be pathogenic (gray histogram), along with the distribution of all missense variants (black line). *j* Distance between variants with different expression levels to AVP in the solved V2R structure.

Line 281. The authors state that they are measuring thermodynamic rescue when they lower the temperature the cells are grown at prior to the screen. Temperature rescue is likely acting through many different mechanisms - some of which may be thermodynamic - others may be

slowing down the process and increasing fidelity. Temperature rescue of CFTR variants has a long history such as: <https://pubmed.ncbi.nlm.nih.gov/1380673/>. And is widely used in structural biology to increase the level of a membrane protein prior to purification. While the temperature rescue DMS is a beautiful experiment, we urge the authors to remove the word thermodynamic in the title and throughout the text and instead use something more descriptive such as 'temperature rescue'.

This comment is addressed in more detail above, but for clarity we repeat the changed text here:

Temperature rescue of V2R variants

In principle, variants that are poorly expressed due to decreased thermodynamic stability should be rescued by incubating the cells expressing the variant library at reduced temperature (Fig. 3a). While reduced temperature could also affect biosynthetic processes that result in increased expression⁵³, we assume that in the majority of the cases rescue is via thermodynamic stabilization¹⁴. We sought to understand what fraction of V2R variants could be rescued by temperature reduction by culturing the cells expressing the V2R variant library at 27°.

Figure 3. The authors spend much of the figure and the corresponding text describing the data processing but not the distributions of the temperature rescued variants. For instance, panel E could be condensed to a single example plot. Overall, we are left with many questions, such as where are these variants in the protein? Are there any trends with physicochemistry, TM#, etc ? W296 is called out as having the “most significantly different” set of residuals but not really discussed, what are some of the other top hits or interesting findings?

We think panel E cannot be condensed to a single example plot- we are trying to illustrate here a multi-step process for finding positions with significantly different residuals compared to the model. Our motivation for calling out W296 is just to show the position with the most significant p-value and show what those data points look like in comparison to all the other data. As a narrative choice, we decided to present the 27° and Tolvaptan rescue analyses first, and then discuss in more detail the top hits and interesting findings in Figure 5 and corresponding text.

Line 314 (related to the previous comment). The authors mention that mutations in rescued are rescue more (6 residues) or less (4) than would be expected from their expression level. This is a bit confusing. What would be expected? Perhaps it would be worth unpacking this a bit more. Further, which ones are more than would be expected and which and which ones are less? Where are these? The authors mention glycosylation sites. Are some glycosylation sites being rescued more while others less? What would this mean? There is a more detailed look at this in supplemental figure 1, but perhaps it would be worth moving the

key bits to the main text?

The “expectation” here is referring to the model- we have updated the text:

For the 27° condition, out of 371 positions, only four positions are rescued less than expected, while six are rescued more than expected based on the model, (FDR=0.1, Fig. 3g-h).

While we only briefly mention the significant sites at this point, we go on to more closely examine them in Figure 5 and associated text. In particular, we show that the O-glycosylation sites are rescued less than expected, while actually one of the N-glycosylation sites is rescued more than expected. It is not clear why this should be, so we simply describe the phenomenon in the text:

Both O-glycosylation sites (S5 and T6) are rescued less than expected, but S24 is actually rescued more than expected, suggesting that temperature reduction can compensate the N-glycosylation defect.

Supplementary figure 1F. Include the mentioned scale bar from fig 1h.

Good suggestion; done.

Line 341 the authors mention that Tolvaptan has a stronger effect than temperature at rescuing cells and reference S Fig3b. While it looks like there is a slight effect it's hard to tell how strong this effect is. I would recommend switching the Plots of supplemental Figure 3a-b to being stacked vs overlaid and reporting numbers and percentages of cells in these populations. That would make it easier to judge this claim but also to see what the increased enrichment is.

Good suggestion- we have now added the percent of cells shifting from low- to high-expressing peaks (Supplemental Figure 3a) and added this information to the text callout as well:

Compared to 27°, the Tolvaptan condition has a stronger rescue effect, with more cells shifting from low to high expression (11% of cells shifting in Tolvaptan compared with 5.3% at 27°, Supplemental Fig. 3b).

Line 366 the authors discuss that there are more positions that have specific interactions w/ Tolvaptan than temperature (32 vs 10). Are all of the 10 shared? If not, why? I think this is the coolest part of the paper and is worth discussing. This demonstrates the power of this systematic approach - to describe specific effects across ALL positions - and is what sets this paper apart from prior studies. Discussing this more would illustrate the strength of the approach more explicitly and could lead to meaningful insights for the membrane protein folding field.

Thanks for the suggestion. We have now included a new discussion of these residues, as well as a comparison of Tolvaptan rescue and 27° rescue:

Of the 10 outlier residues for 27° rescue, 6 are also outliers for Tolvaptan: S5, T6, S167^{4x53}, are rescued less than expected in both cases, and L336^{8x54} is rescued more than expected in both cases. Interestingly, while S24 and Y205^{5x39} are rescued more than expected in 27°, they are actually less rescued than expected in Tolvaptan condition. Overall, though, rescue in the two conditions is correlated ($\rho = 0.49$, Supplemental Fig. 3i).

Figure 4H-I, similar to the comment on the previous figure, S171 is called out in the figure but not really discussed in the text with this section, or even in the structural interpretation with figure 5. There is another position in the top left of the plot adjacent to 171, what is this? What about the other significant positions? A little bit of unpacking might be needed here.

As mentioned above, pulling out S171 is a way to enable the reader to get an immediate, intuitive feeling for what these outlier residues look like. Without these pulled out plots, it might leave the reader wondering what one of these significant outlier positions actually looks like. All of the significant residues are identified on the structure in Figure 5.

Figure 5 and associated text. The authors should mention what the residue superscript means, the GPCRdb/BW numbering system is a bit of a niche thing for GPCRs and not always immediately understood by a broad audience.

Thanks for catching this- we forgot to include a callout to the superscripts! We have included now when describing Figure 5:

Indeed, a study employing molecular dynamics and site-directed mutagenesis⁵⁸ determined that the most important residues for Tolvaptan antagonism are M123^{3x36}, F178^{3x36}, Y205^{5x39}, V206^{5x40}, and F287^{6x51}, of which all but F178^{3x36} (GPCRdb numbering in superscript) are rescued less than expected by Tolvaptan.

Figure 5. We'd encourage the authors to find a new color instead of yellow to highlight residues, it is challenging to read/see this in both print and on a computer screen.

Thanks for pointing this out: we have changed the highlight color to gold and changed the lighting representation of these structural figures, which makes the outlines easier to see.

Line 390. The authors describe the effects of the highly conserved DRY motif. A brief sentence describing the motif and it's role could be helpful for broad readability. There have been a number of GPCR DMS studies conducted including several with surface expression readouts. It may be worth specifically discussing whether mutational effects to this or other canonical GPCR motifs/microswitches generally alter expression.

Related to figure 5b3, the authors suggest that some of these residues are rescued more than expected and perhaps alter signaling and or internalization. Are these residues thought to interface with the G protein? The internalization aspect is also interesting. We'd be curious to know how the authors might interpret any interplay of internalization with the surface expression measurements, especially in the context of variants that might increase basal signaling activity of the receptor.

We agree that explaining a bit the function of DRY motif would be helpful for readability. We have added text:

The E/DRY motif is well conserved among class A GPCRs and plays an important role in stabilizing the inactive state; mutations in this motif cause constitutive activity in various GPCRs⁶³, and similarly mutations at R137^{3x50} in V2R are known to cause constitutive signaling activity⁴⁷,

While we agree that a survey of mutational effects at conserved motifs would indeed be interesting, we leave that to future work.

Line 392-398. The authors describe mutation effects in sites that would bias the ensemble in different ways that could effect drug binding. This is premature and doesn't have a good experimental basis. For instance, backing up these claims would likely need to have experimental measurements of dynamics such as smFRET or MD simulations which have not been done on these mutants. Perhaps just describe the effects in the context of the available knowledge and not reach towards a model of receptor dynamics.

We appreciate the concern that there isn't sufficient evidence to invoke a model of receptor dynamics. While the rescue behavior of most variants can be explained simply (by additive effects of the mutation and the drug binding), in the cases where mutations are rescued to a level greater or lesser than the model predicts, we do think it makes sense to propose some mechanistic explanation. We emphasize the use of qualifying language in proposing these alternative interpretations, and also add a sentence emphasizing the speculative nature of these interpretations:

*Mutations at R137^{3x50} cause constitutive signaling activity⁴⁶, so this cluster of mutations **might** render Tolvaptan less effective by biasing the receptor to the active conformation. An examination of the sites where variants are rescued more than expected highlights a cluster of residues at the intracellular interface of the receptor (Fig. 5b3). Mutations at these sites could **potentially** affect signaling and/or internalization; PC stabilization of the inactive state **might** therefore have an exaggerated effect here. **Further mechanistic studies would be required to understand the behavior of mutations at these sites.***

Line 420. The authors claim temperature rescue proves that variants are disrupting 'thermodynamic instability'. As described throughout this

review - temperature rescue is complex and there is a deep literature on the effects of mutations. We recommend the authors contextualize the results with existing literature. The results here are completely in line with expected results from the field and the way this discussion section is written it sounds like none of these studies have previously been done.

This critique is discussed in more detail above- at the specified lines we have made the change:

*First, we show that over half of known loss-of-function variants are poorly expressed, and use temperature reduction to show that the vast majority of variants lose expression as a result of **presumed** thermodynamic instability.*

Line 429. The authors state that in contrast with prior studies on CFTR and Rhodopsin that in this work they find ‘universal rescue’ as if it disproves previous models of chaperone function. These are different proteins and in the context of the CFTR studies there were multiple chaperones to compare chaperones effects directly. While a beautiful study, this study does not disprove existing models on a completely different protein. This should be acknowledged more explicitly in the discussion.

We agree, and this issue was raised by other reviewers. We made these changes to hopefully add better context to interpret the previous studies:

*Previous low-throughput studies mostly tested the efficacy of PCs for a small number of variants, with the failed rescue of individual variants leading to suggestions of widespread idiosyncratic effects between PCs and variants^{32–35}. More systematic efforts testing hundreds of variants in rhodopsin^{14,68} and CFTR⁶⁹ demonstrated rescue of many variants, but with the authors also suggesting substantial region-specific differences in rescue. **The rhodopsin investigations emphasized differences in rescue between mutations in transmembrane helices 2 and 7¹⁴, but subsequently found that 67 out of 69 reduced expression retinopathy variants had measurably increased expression in the presence of a PC⁶⁸, consistent with the findings here. CFTR has multiple folding domains⁷⁰, and PCs were found to be most effective for variants in close proximity to the PC binding site⁶⁹. Class A GPCRs then, as single-domain proteins, may represent especially amenable targets for PCs.***

Line 440. The authors describe the energetic effects of binding a drug as overcoming the effects of mutants. However, earlier the authors describe the effects of mutants in biasing the ensemble. We agree more with this framing rather than the ensemble based explanation which is partially why we caution the interpretation that mutants alter the conformational ensemble.

Thanks for pointing this out, and this concern is also addressed above. In brief, we are invoking the first model (energetic effects of binding a drug as additively overcoming the effects of mutants) for the vast majority of variants. It is only for those residues that show significantly different rescue from the model that we propose alternative explanations. The residues in Figure 5b1 seem to be clearly affecting Tolvaptan binding, while it is less clear what the residues in 5b2 and 5b3 are doing. We emphasize the use of qualifying language in proposing these alternative interpretations, and also add a sentence emphasizing the speculative nature of these interpretations:

*Mutations at R137^{3x50} cause constitutive signaling activity⁴⁶, so this cluster of mutations **might** render Tolvaptan less effective by biasing the receptor to the active conformation. An examination of the sites where variants are rescued more than expected highlights a cluster of residues at the intracellular interface of the receptor (Fig. 5b3). Mutations at these sites could **potentially** affect signaling and/or internalization; PC stabilization of the inactive state **might** therefore have an exaggerated effect here. **Further mechanistic studies would be required to understand the behavior of mutations at these sites.***

Line 442. The authors describe that chaperones may rescue variants in a more general manner than was previously considered. Pharmacochaperones are widely regarded as a method to rescue expression in proteins, and many companies are currently chasing making them for many membrane protein associated diseases. We don't think this framing is overstating the novelty of the finding and does not accurately represents the state of the field.

We appreciate that many companies are pursuing stabilizers as drug modalities and indeed think it is a very good idea. This does not change the fact though that the literature is devoid of demonstrations of near-universal PC rescue.

Line 448 the authors try to contextualize the results on drugs as the mostly additive effects of mutants. This section will be non-obvious to the vast majority of membrane protein people who will be interested in this paper. We encourage the authors to unpack these ideas a bit. What ideas are similar? Why will the effects be similar? How does the law of mass action apply here? Overall, this paragraph is confusing and complex and we recommend unpacking the ideas and explaining them more.

Thanks for the suggestion, we added this sentence to explain the logic a bit more:

Mutations, like small molecule binding, can be viewed as perturbations to a protein that lead to free energy changes. We believe, therefore, that many small molecules binding to proteins with sufficient free energy will behave as general or universal PCs.

Line 464. The authors refer to similar experiments as a high throughput protein abundance selection experiments and in the next sentence

describes that as shown in this sentence these assays can be useful in exploring chaperones. This is a bit confusing as the prior studies were yeast-based growth based selections whereas this is a surface expression screen. Please adjust the language accordingly.

The similar experiments cited here are *in vitro* display, in human cells, or in yeast cells. We agree that the wording of the second sentence is confusing. We have changed the text to read:

*High-throughput protein abundance selection assays have now been developed for many different classes of protein^{17,71,72}. **In an analogous way to the present study**, these assays **could** be used to rapidly quantify the efficacy of PCs across all variants in a protein to prioritize broadly effective PCs.*

Last paragraph. The authors try to extend that because they find a single ligand rescues nearly all loss of surface expression mutants than any ligand that binds to the receptor could rescue variants. This is an over-reach. If they wanted to make that claim they would need to show many new ligands that bind (including structures or SPR or something like that) all over the receptor that they rescue the same mutants. As the authors have shown the results for a single previously characterized ligand... While the ideas here are interesting, they are not backed up by the manuscript.

We think that the data presented in this manuscript, combined with the interpretation laid out in the Discussion (law of mass action > small molecule binding to native state stabilizes > free energy perturbations likely to be additive) does support our claim that *many* small molecules will behave as general PCs:

We believe, therefore, that many small molecules binding to proteins with sufficient free energy will behave as general or universal PCs.

Data availability: We strongly encourage the authors to upload the DMS data to MaveDB.

We have already made all data available on zenodo, and upon acceptance / finalization of the paper we will upload to MaveDB.

By Matthew Howard and Willow Coyote-Maestas

Reviewer #3 (Remarks to the Author):

Protein misfolding has been implicated in the molecular basis of numerous genetic diseases, and the recent development of small molecule pharmacochaperones that suppress misfolding reactions has revolutionized the treatment of several such diseases. In theory, mechanistic basis for the activity of these molecules is quite simple- ligands that selectively bind to the native fold should enhance folding via mass action. However, in the ~ 20 years since the conceptual development of such molecules by Jeff Kelly and colleagues, it has become quite clear that the stabilization generated by such molecules can play out in nuanced ways in the context of the cell. For instance, while pharmacochaperones that target the CFTR protein have revolutionized the treatment of cystic fibrosis, the best molecules and combinatorial therapies that are currently available in the clinic are only effective towards certain clinical CFTR variants. Ongoing work to understand how these molecules do or don't rescue misfolded variants and how we can maximize their efficacy towards diverse clinical genotypes is an important challenge in precision medicine. In this work, Mighell & Lehner investigate how an existing drug that acts as a pharmacochaperone (Tolvaptan) modulates the expression of a comprehensive library of ~7k variants of the vasopressin-2 receptor (V2R), the misfolding of which causes nephrogenic diabetes insipidus (NDI). While several previous reports have demonstrated that select misfolded clinical V2R variants can be rescued by pharmacochaperones, this work provides a more comprehensive profile of the expression pattern, temperature sensitivity, and Tolvaptan response of thousands of random missense variants. Their findings show that about half of the clinical NDI variants reduce V2R expression, and that most of those can be rescued, to some extent, by Tolvaptan. This is a wonderful, impressive, and technically sound data set, and the manuscript reports a variety of interesting findings. However, while the writing is quite clear, I find the overarching interpretations to be overstated and several of their conclusions to be somewhat misplaced. There are also several aspects of the manuscript that could benefit from clarification. I have summarized my suggestions as follows:

We thank the referee for their enthusiasm and very constructive suggestions.

Conceptual Issues:

-Overall, the manuscript seems to imply that the idea that pharmacochaperone activity should be generalizable to a variety of misfolded variants is a novel feature of this manuscript. For instance, the authors state "Here we use V2R as a model system to directly test the hypothesis that PCs can rescue destabilizing mutants across the structure of a complex human protein." I appreciate that there are a fair number of cellular biologists that view the effects of these compounds from a phenomenological/ empirical viewpoint, and think these points are worth re-stating in the plainest of language until this perspective is widely appreciated. Nevertheless, I think many biochemists and biophysicists

within the proteostasis community have made these points many times over the years (see doi: 10.1146/annurev.biochem.052308.114844 and 10.1016/j.sbi.2021.11.009). In fact, in light of the physical arguments made by these authors and many others, the mere fact that all variants do not respond to such compounds is perhaps the most interesting part. Based on these considerations, I think the fact that the authors have described this paper as a proof of concept for the fact that some pharmacochaperones can rescue lots of mutants seems a bit odd and inappropriate. It doesn't stress the most interesting parts of the manuscript either, in my opinion.

We appreciate the insight offered by the reviewer. In our reading of the literature, (as you point out) we found many contradictory assertions that stabilizer action could be specific and idiosyncratic, or that it could be general and simply described by the law of mass action. In the introduction, we briefly present this dichotomy. To more clearly lay out the dichotomy and existing ideas we have added a callout to the law of mass action and the recommended citation, as well as specifying that some PCs may have a variant-specific mechanism, while some may be much more general:

*Depending on the mechanism of action, **some** PCs could have high specificity, rescuing only subsets of pathogenic variants localized in particular regions of the protein^{13,14}. Alternatively, **some** PCs could behave simply, in accordance with the law of mass action, and have largely nonspecific stabilizing effects that offset the destabilization by most variants in a protein, wherever they are located^{12,15}.*

We agree that the fact that not all variants respond is interesting: in fact, this is why we included the residuals/outliers analyses in Figure 3, 4, and 5. We try to emphasize that the non-rescued positions are in the Tolvaptan binding site, or other functionally important sites in the protein. Of course, any study can be framed in different ways; while the reviewer may find detailed analysis of membrane protein biogenesis mechanisms the most interesting, we decided that the nearly-completely effective nature of Tolvaptan rescue would be the most interesting result to the broadest audience.

-As the authors have noted, several other pharmacochaperones have been characterized against variants of several other target proteins and few, if any, have demonstrated such a broad efficacy towards such a diverse range of mutants as Tolvaptan. The authors imply in the discussion that the previous studies on this topic have somehow missed the point because they didn't look at enough mutants, noting "the failed rescue of individual variants (led) to suggestions of widespread idiosyncratic effects between PCs and variants." But this seems like strange logic. The fact that these authors have identified such widespread rescue for a different protein with a different pharmacochaperone doesn't mean the previous studies are somehow invalid or misguided. It is certainly possible (and perhaps likely) that if previous investigations of other targets had profiled thousands more variants, they may have identified significantly more mutants that can be corrected by these other

pharmacochaperones. However, collecting more measurements for random mutations and using them to train new models won't change the fact that the "idiosyncratic" clinical mutations focused on by these investigations don't respond to the drugs in question- the most important pharmacological issue. Given the divergent observations the authors have made for random mutations relative to more focused investigations of disease mutants, it seems more appropriate to consider whether this is a unique property of this target receptor relative to other targets. Or perhaps the disease mutants in V2R differ from those of CFTR or other proteins? Or perhaps there is some feature of this specific compounds that endows it with broader efficacy relative to others (i.e. large binding energy). From my experience with CFTR modulators, I can tell you this later point seems most plausible, as different ligands with similar binding energy can have strikingly different effects- an observation that cannot be explained by thermodynamics. Indeed, cells are not at equilibrium, and most biochemical processes are subject to kinetic control, including the coupling between folding and trafficking in the ER. Energetics are highly useful, but often do not tell the whole story. I feel like this is a more useful discussion to have for the field.

We agree with the reviewer that our framing made it sound like previous studies were somehow wrong. We have tried to address this and have taken the opportunity to expand the discussion around previous findings in the discussion

Previous low-throughput studies mostly tested the efficacy of PCs for a small number of variants, with the failed rescue of individual variants leading to suggestions of widespread idiosyncratic effects between PCs and variants³²⁻³⁵. More systematic efforts testing hundreds of variants in rhodopsin^{14,68} and CFTR⁶⁹ demonstrated rescue of many variants, but with the authors also suggesting substantial region-specific differences in rescue. The rhodopsin investigations emphasized differences in rescue between mutations in transmembrane helices 2 and 7¹⁴, but subsequently found that 67 out of 69 reduced expression retinopathy variants had measurably increased expression in the presence of a PC⁶⁸, consistent with the findings here. CFTR has multiple folding domains⁷⁰, and PCs were found to be most effective for variants in close proximity to the PC binding site⁶⁹. Class A GPCRs then, as single-domain proteins, may represent amenable targets for PCs.

-The authors state: "To maximize the effectiveness of PC therapy though, it is necessary to identify the mechanism of all pathogenic variation for a given protein, as well as the response to PC. These data do not currently exist for any protein" I think this is an overly strong statement. I think recent work on CFTR is perhaps the most comprehensive to date (doi: 10.1016/j.jcf.2024.02.006). While this study does not report 7k random variants, it includes all 655 currently known pathogenic mutations and multiple pharmacochaperone treatments. Importantly, this investigation also uses techniques that are accepted by the US FDA for the expansion of the therapeutic label of these drugs, which is something that DMS

unfortunately cannot be used for currently. It is also worth noting that a comprehensive (albeit small) library of pathogenic rhodopsin mutants has been used to profile five different pharmacochaperones to date (doi: 10.1016/j.jbc.2022.102266, 10.1093/hmg/ddac125, and 10.1371/journal.pbio.3002932). Though these libraries are an order of magnitude smaller than the library of random variants characterized in this work, they are arguably more complete from the perspective of clinical genetics, as they include indels that are found in the CF and RP patient population. The omission of these variants from missense libraries is not a trivial issue, as indels are quite common in the clinic and can be highly penetrant (ie $\Delta F508$ CFTR).

This is a good point and we agree. We have omitted the sentence “these data do not currently exist for any protein”.

-The authors state “the stabilization conferred by small molecule binding should be largely independent of where a small molecule binds and where a mutation is located, provided the compound specifically binds the native folded state.” While we agree this should be the case, the authors have noted several cases in the literature that show that empirically, this is not actually the case (see refs 14, 36, and 68). Again, these observations are not invalid simply because a larger number of other random mutants were not included in these libraries. The regiospecific variations and the distinct effects of pharmacochaperones that bind to different sites would still be there, regardless of the library size.

We agree that this point needs to be made more clearly. As noted before, we have now changed the discussion to explicitly discuss the possibility of PCs being differentially effective across folding domains. Changed text repeated here for clarity:

*Previous low-throughput studies mostly tested the efficacy of PCs for a small number of variants, with the failed rescue of individual variants leading to suggestions of widespread idiosyncratic effects between PCs and variants³²⁻³⁵. More systematic efforts testing hundreds of variants in rhodopsin^{14,68} and CFTR⁶⁹ demonstrated rescue of many variants, but with the authors also suggesting substantial region-specific differences in rescue. **The rhodopsin investigations emphasized differences in rescue between mutations in transmembrane helices 2 and 7¹⁴, but subsequently found that 67 out of 69 reduced expression retinopathy variants had measurably increased expression in the presence of a PC⁶⁸, consistent with the findings here. CFTR has multiple folding domains⁷⁰, and PCs were found to be most effective for variants in close proximity to the PC binding site⁶⁹. Class A GPCRs then, as single-domain proteins, may represent amenable targets for PCs.***

-The authors note “Namely, the magnitude of rescue is greatest for variants with starting free energy values in the steepest part of the free energy- expression level curve (Fig. 3d).” I agree with the authors

interpretation of this observation. However, the authors should note that similar observations have been previously made for both CFTR and rhodopsin variants (see Refs 67 and 68). The authors cited these papers to note they used a relatively small library, but failed to mention that both of these studies made similar observations and offered this same explanation previously. It is instead offered as a novel finding made in this study. It is worth noting that the sensitivity of the moderately expressed variants seems to be an emergent phenomenon that appears to apply to most targets. I should also note that, had the authors applied the thermodynamic bounding analysis outlined in refs 67 and 68, I suspect they would find the trends in Tolvaptam response are relatively similar to those observed for CFTR and rhodopsin- many respond but not to the extent that the thermodynamic effects of ligand binding would suggest they should. This phenomenon is not so much a yes/ no question, as the authors manuscript suggests, but rather more a question of how much. Discussing which aspects of this phenomenon are the same or instead differ across target proteins is much more useful than litigating methodological shortcomings.

We agree that we should explicitly call out that this phenomenon has been observed before. We make these changes:

Namely, as has been observed before in similar experimental paradigms,^{54,55} the magnitude of rescue is greatest for variants with starting free energy values in the steepest part of the free energy-expression level curve (Fig. 3d).

Technical Issues:

In their description of how they calculated variant scores, the authors note that read counts were “multiplied by the geometric mean fluorescence value associated with each bin.” Could the authors please explain why they chose to use the geometric mean instead of the arithmetic mean? I am aware that many in the flow cytometry community still use geometric means. However, it is my understanding that this is a relic from a bygone era- geometric mean was previously used when cytometers had few fluorescence channels and intensity measurements were collected using log scale bins to avoid compressing dynamic range. The geometric mean of log data transforms it into a linear mean, which overcame the limitation of collecting logarithmic data. But these days, most cytometers have 256k channels and collect data using linear bins. Is there a reason for using a geometric mean in this case? Will it improve variant scoring? I am genuinely curious.

Since the distribution of fluorescence values is log-distributed, we reasoned that the geometric mean fluorescence value of each bin would better represent the cell population of that bin than the arithmetic mean. However, as this value is simply a way to scale the data, we expected that there shouldn't be a substantive difference between using either of these values. To be sure of this though, we re-ran the scoring analysis using arithmetic mean and compared

scores between arithmetic and geometric mean and they were correlated at a level of Pearson's $r = 0.999$.

-The authors provide little detail on how variant scores are binned into the poorly expressed, moderately expressed, and well-expressed classifications. Were the boundaries indexed to some reference (ie WT or a misfolded mutant)? I recognize that this is an arbitrary detail that will likely not affect the outcome, but I know that I will not be the only reader to consider this issue. It might be good to add a brief statement on the criterion and/ or provide a supplemental analysis showing the conclusion is relatively insensitive to the positions of the bins.

We use the distribution of synonymous and nonsense variants to define these three categories. In Results section "Massively parallel measurement of V2R surface expression" we say:

We used the upper 95th percentile of truncation scores (0.35) and the lower 95th percentile of synonymous wild-type scores (0.825) to categorize missense variants as well expressed (3,415 variants), moderately expressed (1,772 variants), or poorly expressed (1,025 variants, Fig. 1b).

These thresholds are also represented as dashed lines in Figure 1b.

-By DMS standards, the authors note admirable precision between replicates in the supplement. However, the effects of Tolvaptam on many variants is relatively subtle. In this regard, the authors use of a larger library has its own tradeoffs, such as lower signal to noise due to dilute sampling. In evaluating the reported scores, it seems that for many variants (if not most), the difference between the scores in the presence and absence of drug is small relative to the SEM of each measurement. There is little discussion of the precision of the shifts and, indeed, the variants of interest in Fig. 4i depict error bars without even noting what they correspond to. The authors should provide an analysis, complete with error propagation, that states how many of the observed shifts are statistically robust. I don't mean to quibble, but such considerations will be important for the design of future DMS screening approaches, where the comparison of subtle drug effects on specific variants could be highly consequential for decision making that may occur during the drug development process.

Thanks for spotting the omission in Fig. 4i- we have updated the figure legend to include:

Error bars represent the standard error of the mean.

It is a good idea to provide an alternate estimation of the fraction of variants rescued. We have included a new figure panel (Supplemental figure 3h) that compares Control expression level with magnitude of tolvaptan rescue. And, we have added this text to the main text:

As an alternative estimation of the extent of rescue, we identified those variants with Tolvaptan rescue magnitude (i.e. Tolvaptan expression – control expression) greater than the 95% confidence interval of Tolvaptan expression. While the rescue magnitude for well-expressed variants is expected to be small, when considering variants with low control expression (<0.6), 88.4% of variants are significantly rescued (Supplemental Fig. 3h).

Overall, I would like to re-affirm that this is an excellent study that deserves to be highlighted in this prestigious journal. I just think the authors should first make an earnest effort to bridge this study with other emerging observations rather than focusing on the potential shortcomings of previous methodologies. There is more to be learned from the nuance within their data than can be learned from glossing over the outliers. I genuinely hope the authors find these comments useful. With improvements, this is sure to be a landmark paper.

Thank you for these statements.

Reviewer #1:

Remarks to the Author:

I have read the revised manuscript in full, and only have a few remaining minor comments:

1) Lines 173 to 177: The manuscript mentions "The predicted free energy change (ΔG) of membrane integration for these helices is relatively unfavorable", and "TM7 is predicted to be inefficiently incorporated in the membrane", but it's hard to put too much weight into these statements without seeing some corresponding data / values.

It's a good point that these statements are difficult to assess without values. So, we have included values (changed text in bold):

The predicted free energy change (ΔG) of membrane integration⁴² for these helices is relatively unfavorable (**median predicted ΔG of 3.1 and 3.7 for TM2 and TM3, respectively**) suggesting that mutations here could further compromise an already inefficient process. In contrast, TM7 is predicted to be inefficiently incorporated in the membrane (**median predicted ΔG of 4.2**), while mutations here are well tolerated (Fig. 1f).

2) Lines 314 (and elsewhere, later on): Should Lowess be capitalized to LOWESS, since it's an acronym?

Yes thank you for catching this; we have changes all occurrences of "Lowess" to "LOWESS"

3) Line 383: The GPCRdb superscripted numbering is first used here, but first defined on line 408. I suggest moving the parenthetical definition up to this line.

Thank you for catching this! We have moved the callout explaining GPCRdb numbering.

Reviewer #2:

Remarks to the Author:

The authors did a fantastic job addressing the all the concerns I had either by making adjustments in the text, the figures, or explaining their rationale. Kudos for a beautiful paper!

Willow Coyote-Maestas and Matthew K. Howard

Reviewer #3:

Remarks to the Author:

The authors have done an adequate job of addressing the many lengthy requests made by the reviewers. I am grateful for their careful consideration and have no further requests.